# Better Safe Than Sorry: Preventing Delusive Adversaries with Adversarial Training

Lue Tao[1,2]    Lei Feng[3]    Jinfeng Yi[4]    Sheng-Jun Huang[1,2]    Songcan Chen[1,2][†]

[1]College of Computer Science and Technology, Nanjing University of Aeronautics and Astronautics
[2]MIIT Key Laboratory of Pattern Analysis and Machine Intelligence
[3]College of Computer Science, Chongqing University
[4]JD AI Research

## Abstract

*Delusive attacks* aim to substantially deteriorate the test accuracy of the learning model by *slightly* perturbing the *features* of correctly labeled training examples. By formalizing this malicious attack as finding the worst-case training data within a specific $\infty$-Wasserstein ball, we show that minimizing adversarial risk on the *perturbed data* is equivalent to optimizing an upper bound of natural risk on the *original data*. This implies that adversarial training can serve as a *principled* defense against delusive attacks. Thus, the test accuracy decreased by delusive attacks can be largely recovered by adversarial training. To further understand the internal mechanism of the defense, we disclose that adversarial training can resist the delusive perturbations by preventing the learner from overly relying on non-robust features in a natural setting. Finally, we complement our theoretical findings with a set of experiments on popular benchmark datasets, which show that the defense withstands six different practical attacks. Both theoretical and empirical results vote for adversarial training when confronted with delusive adversaries.

## 1   Introduction

Although machine learning (ML) models have achieved superior performance on many challenging tasks, their performance can be largely deteriorated when the training and test distributions are different. For instance, standard models are prone to make mistakes on the adversarial examples that are considered as worst-case data at *test* time [10, 113]. Compared with that, a more threatening and easily overlooked threat is the malicious perturbations at *training* time, i.e., the *delusive attacks* [82] that aim to maximize test error by slightly perturbing the correctly labeled training examples [6, 7].

In the era of big data, many practitioners collect training data from untrusted sources where delusive adversaries may exist. In particular, many companies are scraping large datasets from unknown users or public websites for commercial use. For example, Kaspersky Lab, a leading antivirus company, has been accused of poisoning competing products [7]. Although they denied any wrongdoings and clarified the false rumors, one can still imagine the disastrous consequences if that really happens in the security-critical applications. Furthermore, a recent survey of 28 organizations found that these industry practitioners are obviously more afraid of data poisoning than other threats from adversarial ML [64]. In a nutshell, delusive attacks has become a realistic and horrible threat to practitioners.

Recently, Feng et al. [32] showed for the first time that delusive attacks are feasible for deep networks, by proposing "training-time adversarial data" that can significantly deteriorate model performance on clean test data. However, how to design learning algorithms that are robust to delusive attacks still remains an open question due to several crucial challenges [82, 32]. First of all, delusive attacks

---

[†]Corresponding author: Songcan Chen <s.chen@nuaa.edu.cn>.

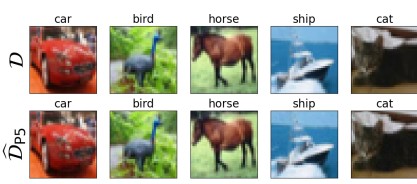
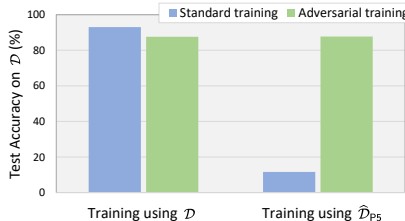

Figure 1: An illustration of delusive attacks and adversarial training. **Left**: Random samples from the CIFAR-10 [63] training set: the original training set $\mathcal{D}$; the perturbed training set $\widehat{\mathcal{D}}_{\mathsf{P5}}$, generated using the P5 attack described in Section 4. **Right**: Natural accuracy evaluated on clean test set for models trained with: *i)* standard training on $\mathcal{D}$; *ii)* adversarial training on $\mathcal{D}$; *iii)* standard training on $\widehat{\mathcal{D}}_{\mathsf{P5}}$; *iv)* adversarial training on $\widehat{\mathcal{D}}_{\mathsf{P5}}$. While standard training on $\widehat{\mathcal{D}}_{\mathsf{P5}}$ incurs poor generalization performance on $\mathcal{D}$, adversarial training can help a lot. Details are deferred to Section 5.1.

cannot be avoided by standard data cleaning [59], since they does not require mislabeling, and the perturbed examples will maintain their malice even when they are correctly labeled by experts. In addition, even if the perturbed examples could be distinguished by some detection techniques, it is wasteful to filter out these correctly labeled examples, considering that deep models are data-hungry. In an extreme case where all examples in the training set are perturbed by a delusive adversary, there will leave no training examples after the filtering stage, thus the learning process is still obstructed. Given these challenges, we aim to examine the following question in this study: *Is it possible to defend against delusive attacks without abandoning the perturbed examples?*

In this work, we provide an affirmative answer to this question. We first formulate the task of delusive attacks as finding the worst-case data at training time within a specific $\infty$-Wasserstein ball that prevents label changes (Section 2). By doing so, we find that minimizing the *adversarial risk* on the *perturbed data* is equivalent to optimizing an upper bound of natural risk on the *original data* (Section 3.1). This implies that *adversarial training* [44, 74] on the perturbed training examples can maximize the natural accuracy on the clean examples. Further, we disclose that adversarial training can resist the delusive perturbations by preventing the learner from overly relying on the non-robust features (that are predictive, yet brittle or incomprehensible to humans) in a simple and natural setting. Specifically, two opposite perturbation directions are studied, and adversarial training helps in both cases with different mechanisms (Section 3.2). All these evidences suggest that adversarial training is a promising solution to defend against delusive attacks.

Importantly, our findings widen the scope of application of adversarial training, which was only considered as a principled defense method against test-time adversarial examples [74, 21]. Note that adversarial training usually leads to a drop in natural accuracy [119]. This makes it less practical in many real-world applications where test-time attacks are rare and a high accuracy on clean test data is required [65]. However, this study shows that adversarial training can also defend against a more threatening and invisible threat called delusive adversaries (see Figure 1 for an illustration). We believe that adversarial training will be more widely used in practical applications in the future.

Finally, we present five practical attacks to empirically evaluate the proposed defense (Section 4). Extensive experiments on various datasets (CIFAR-10, SVHN, and a subset of ImageNet) and tasks (supervised learning, self-supervised learning, and overcoming simplicity bias) demonstrate the effectiveness and versatility of adversarial training, which significantly mitigates the destructiveness of various delusive attacks (Section 5). Our main contributions are summarized as follows:

- **Formulation of delusive attacks.** We provide the first attempt to formulate delusive attacks using the $\infty$-Wasserstein distance. This formulation is novel and general, and can cover the formulation of the attack proposed by Feng et al. [32].
- **The principled defense.** Equipped with the novel characterization of delusive attacks, we are able to show that, for the first time, adversarial training can serve as a *principled* defense against delusive attacks with theoretical guarantee (Theorem 1).
- **Internal Mechanisms.** We further disclose the internal mechanisms of the defense in a popular mixture-Gaussian setting (Theorem 2 and Theorem 3).
- **Empirical evidences.** We complement our theoretical findings with extensive experiments across a wide range of datasets and tasks.

## 2 Preliminaries

In this section, we introduce some notations and the main ideas we build upon: natural risk, adversarial risk, Wasserstein distance, and delusive attacks.

**Notation.** Consider a classification task with data $(\boldsymbol{x}, y) \in \mathcal{X} \times \mathcal{Y}$ from an underlying distribution $\mathcal{D}$. We seek to learn a model $f : \mathcal{X} \to \mathcal{Y}$ by minimizing a loss function $\ell(f(\boldsymbol{x}), y)$. Let $\Delta : \mathcal{X} \times \mathcal{X} \to \mathbb{R}$ be some distance metric. Let $\mathcal{B}(\boldsymbol{x}, \epsilon, \Delta) = \{\boldsymbol{x}' \in \mathcal{X} : \Delta(\boldsymbol{x}, \boldsymbol{x}') \leq \epsilon\}$ be the ball around $\boldsymbol{x}$ with radius $\epsilon$. When $\Delta$ is free of context, we simply write $\mathcal{B}(\boldsymbol{x}, \epsilon, \Delta) = \mathcal{B}(\boldsymbol{x}, \epsilon)$. Throughout the paper, the adversary is allowed to perturb only the inputs, not the labels. Thus, similar to Sinha et al. [109], we define the cost function $c : \mathcal{Z} \times \mathcal{Z} \to \mathbb{R} \cup \{\infty\}$ by $c(\boldsymbol{z}, \boldsymbol{z}') = \Delta(\boldsymbol{x}, \boldsymbol{x}') + \infty \cdot \mathbf{1}\{y \neq y'\}$, where $\boldsymbol{z} = (\boldsymbol{x}, y)$ and $\mathcal{Z}$ is the set of possible values for $(\boldsymbol{x}, y)$. Denote by $\mathcal{P}(\mathcal{Z})$ the set of all probability measures on $\mathcal{Z}$. For any $\boldsymbol{\mu} \in \mathbb{R}^d$ and positive definite matrix $\boldsymbol{\Sigma} \in \mathbb{R}^{d \times d}$, denote by $\mathcal{N}(\boldsymbol{\mu}, \boldsymbol{\Sigma})$ the $d$-dimensional Gaussian distribution with mean vector $\boldsymbol{\mu}$ and covariance matrix $\boldsymbol{\Sigma}$.

**Natural risk.** Standard training (ST) aims to minimize the natural risk, which is defined as

$$\mathcal{R}_{\mathsf{nat}}(f, \mathcal{D}) = \mathbb{E}_{(\boldsymbol{x}, y) \sim \mathcal{D}} \left[ \ell(f(\boldsymbol{x}), y) \right]. \tag{1}$$

The term "natural accuracy" refers to the accuracy of a model evaluated on the unperturbed data.

**Adversarial risk.** The goal of adversarial training (AT) is to minimize the adversarial risk defined as

$$\mathcal{R}_{\mathsf{adv}}(f, \mathcal{D}) = \mathbb{E}_{(\boldsymbol{x}, y) \sim \mathcal{D}}[\max_{\boldsymbol{x}' \in \mathcal{B}(\boldsymbol{x}, \epsilon)} \ell(f(\boldsymbol{x}'), y)], \tag{2}$$

which is a robust optimization problem that considers the worst-case performance under pointwise perturbations within an $\epsilon$-ball [74]. The main assumption here is that the inputs satisfying $\Delta(\boldsymbol{x}, \boldsymbol{x}') \leq \epsilon$ preserve the label $y$ of the original input $\boldsymbol{x}$.

**Wasserstein distance.** Wasserstein distance is a distance function defined between two probability distributions, which represents the cost of an optimal mass transportation plan. Given two data distributions $\mathcal{D}$ and $\mathcal{D}'$, the *p-th Wasserstein distance*, for any $p \geq 1$, is defined as:

$$\mathrm{W}_p(\mathcal{D}, \mathcal{D}') = (\inf_{\gamma \in \Pi(\mathcal{D}, \mathcal{D}')} \int_{\mathcal{Z} \times \mathcal{Z}} c(\boldsymbol{z}, \boldsymbol{z}')^p d\gamma(\boldsymbol{z}, \boldsymbol{z}'))^{1/p}, \tag{3}$$

where $\Pi(\mathcal{D}, \mathcal{D}')$ is the collection of all probability measures on $\mathcal{Z} \times \mathcal{Z}$ with $\mathcal{D}$ and $\mathcal{D}'$ being the marginals of the first and second factor, respectively. The $\infty$-Wasserstein distance is defined as the limit of $p$-th Wasserstein distance, i.e., $\mathrm{W}_\infty(\mathcal{D}, \mathcal{D}') = \lim_{p \to \infty} \mathrm{W}_p(\mathcal{D}, \mathcal{D}')$. The *p-th Wasserstein ball* with respect to $\mathcal{D}$ and radius $\epsilon \geq 0$ is defined as: $\mathcal{B}_{\mathrm{W}_p}(\mathcal{D}, \epsilon) = \{\mathcal{D}' \in \mathcal{P}(\mathcal{Z}) : \mathrm{W}_p(\mathcal{D}, \mathcal{D}') \leq \epsilon\}$.

**Delusive adversary.** The *attacker* is capable of manipulating the training data, as long as the training data is correctly labeled, to prevent the *defender* from learning an accurate classifier [82]. Following Feng et al. [32], the game between the attacker and the defender proceeds as follows:

- $n$ data points are drawn from $\mathcal{D}$ to produce a clean training dataset $\mathcal{D}_n$.
- The attacker perturbs some inputs $\boldsymbol{x}$ in $\mathcal{D}_n$ by adding small perturbations to produce $\boldsymbol{x}'$ such that $\Delta(\boldsymbol{x}, \boldsymbol{x}') \leq \epsilon$, where $\epsilon$ is a small constant that represents the attacker's budget. The (partially) perturbed inputs and their original labels constitute the perturbed dataset $\widehat{\mathcal{D}}_n$.
- The defender trains on the perturbed dataset $\widehat{\mathcal{D}}_n$ to produce a model, and incurs natural risk.

The attacker's goal is to maximize the natural risk while the defender's task is to minimize it. We then formulate the attacker's goal as the following bi-level optimization problem:

$$\max_{\widehat{\mathcal{D}} \in \mathcal{B}_{\mathrm{W}_\infty}(\mathcal{D}, \epsilon)} \mathcal{R}_{\mathsf{nat}}(f_{\widehat{\mathcal{D}}}, \mathcal{D}), \quad \text{s.t. } f_{\widehat{\mathcal{D}}} = \arg\min_f \mathcal{R}_{\mathsf{nat}}(f, \widehat{\mathcal{D}}). \tag{4}$$

In other words, Eq. (4) is seeking the training data bounded by the $\infty$-Wasserstein ball with radius $\epsilon$, so that the model trained on it has the worst performance on the original distribution.

**Remark 1.** It is worth noting that using the $\infty$-Wasserstein distance to constrain delusive attacks possesses several advantages. Firstly, the cost function $c$ used in Eq. (3) prevents label changes after perturbations since we only consider clean-label attacks. Secondly, our formulation does not restrict the choice of the distance metric $\Delta$ of the input space, thus our theoretical analysis works with any metric, including the $\ell_\infty$ threat model considered in Feng et al. [32]. Finally, the $\infty$-Wasserstein ball is more aligned with adversarial risk than other uncertainty sets [110, 147].

**Remark 2.** Our formulation assumes an underlying distribution that represents the perturbed dataset. This assumption has been widely adopted by existing works [111, 118, 145]. The rationale behind the assumption is that generally, the defender treats the training dataset as an empirical distribution and trains the model on randomly shuffled examples (e.g., training deep networks via stochastic gradient descent). It is also easy to see that our formulation covers the formulation of Feng et al. [32]. On the other hand, this assumption has its limitations. For example, if the defender treats the training examples as sequential data [24], the attacker may utilize the dependence in the sequence to construct perturbations. This situation is beyond the scope of this work, and we leave it as future work.

## 3 Adversarial Training Beats Delusive Adversaries

In this section, we first justify the rationality of adversarial training as a principled defense method against delusive attacks in the *general* case for *any* data distribution. Further, to understand the internal mechanism of the defense, we explicitly explore the space that delusive attacks can exploit in a simple and natural setting. This indicates that adversarial training resists the delusive perturbations by preventing the learner from overly relying on the non-robust features.

### 3.1 Adversarial Risk Bounds Natural Risk

Intuitively, the original training data is close to the data perturbed by delusive attacks, so it should be found in the vicinity of the perturbed data. Thus, training models around the perturbed data can translate to a good generalization on the original data. We make the intuition formal in the following theorem, which indicates that adversarial training on the perturbed data is actually minimizing an upper bound of natural risk on the original data.

**Theorem 1.** *Given a classifier $f : \mathcal{X} \to \mathcal{Y}$, for any data distribution $\mathcal{D}$ and any perturbed distribution $\widehat{\mathcal{D}}$ such that $\widehat{\mathcal{D}} \in \mathcal{B}_{W_\infty}(\mathcal{D}, \epsilon)$, we have*

$$\mathcal{R}_{\mathsf{nat}}(f, \mathcal{D}) \leq \max_{\mathcal{D}' \in \mathcal{B}_{W_\infty}(\widehat{\mathcal{D}}, \epsilon)} \mathcal{R}_{\mathsf{nat}}(f, \mathcal{D}') = \mathcal{R}_{\mathsf{adv}}(f, \widehat{\mathcal{D}}).$$

The proof is provided in Appendix C.1. Theorem 1 suggests that adversarial training is a principled defense method against delusive attacks. Therefore, when our training data is collected from untrusted sources where delusive adversaries may exist, adversarial training can be applied to minimize the desired natural risk. Besides, Theorem 1 also highlights the importance of the budget $\epsilon$. On the one hand, if the defender is overly pessimistic (i.e., the defender's budget is larger than the attacker's budget), the tightness of the upper bound cannot be guaranteed. On the other hand, if the defender is overly optimistic (i.e., the defender's budget is relatively small or even equals to zero), the natural risk on the original data cannot be upper bounded anymore by the adversarial risk. Our experiments in Section 5.1 cover these cases when the attacker's budget is not specified.

### 3.2 Internal Mechanism of the Defense

To further understand the internal mechanism of the defense, in this subsection, we consider a simple and natural setting that allows us to explicitly manipulate the non-robust features. It turns out that, similar to the situation in adversarial examples [119, 56], the model's reliance on non-robust features also allows delusive adversaries to take advantage of it, and adversarial training can resist delusive perturbations by preventing the learner from overly relying on the non-robust features.

As Ilyas et al. [56] has clarified that both robust and non-robust features in data constitute useful signals for standard classification, we are motivated to consider the following binary classification problem on a mixture of two Gaussian distributions $\mathcal{D}$:

$$y \overset{u \cdot a \cdot r}{\sim} \{-1, +1\}, \quad \boldsymbol{x} \sim \mathcal{N}(y \cdot \boldsymbol{\mu}, \sigma^2 \boldsymbol{I}), \tag{5}$$

where $\boldsymbol{\mu} = (1, \eta, \ldots, \eta) \in \mathbb{R}^{d+1}$ is the mean vector which consists of 1 robust feature with center 1 and $d$ non-robust features with corresponding centers $\eta$, similar to the settings in Tsipras et al. [119]. Typically, there are far more non-robust features than robust features (i.e., $d \gg 1$). To restrict the capability of delusive attacks, here we chose the metric function $\Delta(\boldsymbol{x}, \boldsymbol{x}') = \|\boldsymbol{x} - \boldsymbol{x}'\|_\infty$. We assume that the attacker's budget $\epsilon$ satisfies $\epsilon \geq 2\eta$ and $\eta \leq 1/3$, so that the attacker: *i)* can shift

each non-robust feature towards becoming anti-correlated with the correct label; *ii)* cannot shift each non-robust feature to be strongly correlated with the correct label (as strong as the robust feature).

**Delusive attack is easy.** For the sake of illustration, here we choose $\epsilon = 2\eta$ and consider two opposite perturbation directions. One direction is that all non-robust features shift towards $-y$, the other is to shift towards $y$. These settings are chosen for mathematical convenience. The following analysis can be easily adapted to any $\epsilon \geq 2\eta$ and any combinations of the two directions on non-robust features.

Note that the Bayes optimal classifier (i.e., minimization of the natural risk with 0-1 loss) for the distribution $\mathcal{D}$ is $f_{\mathcal{D}}(\boldsymbol{x}) = \text{sign}(\boldsymbol{\mu}^\top \boldsymbol{x})$, which relies on both robust feature and non-robust features. As a result, an $\ell_\infty$-bounded delusive adversary that is only allowed to perturb each non-robust feature by a moderate $\epsilon$ can take advantage of the space of non-robust features. Formally, the original distribution $\mathcal{D}$ can be perturbed to the delusive distribution $\widehat{\mathcal{D}}_1$:

$$y \overset{u.a.r}{\sim} \{-1, +1\}, \quad \boldsymbol{x} \sim \mathcal{N}(y \cdot \widehat{\boldsymbol{\mu}}_1, \sigma^2 \boldsymbol{I}), \tag{6}$$

where $\widehat{\boldsymbol{\mu}}_1 = (1, -\eta, \ldots, -\eta)$ is the shifted mean vector. After perturbation, every non-robust feature is correlated with $-y$, thus the Bayes optimal classifier for $\widehat{\mathcal{D}}_1$ would yield extremely poor performance on $\mathcal{D}$, for $d$ large enough. Another interesting perturbation direction is to strengthen the magnitude of non-robust features. This leads to the delusive distribution $\widehat{\mathcal{D}}_2$:

$$y \overset{u.a.r}{\sim} \{-1, +1\}, \quad \boldsymbol{x} \sim \mathcal{N}(y \cdot \widehat{\boldsymbol{\mu}}_2, \sigma^2 \boldsymbol{I}), \tag{7}$$

where $\widehat{\boldsymbol{\mu}}_2 = (1, 3\eta, \ldots, 3\eta)$ is the shifted mean vector. Then, the Bayes optimal classifier for $\widehat{\mathcal{D}}_2$ will overly rely on the non-robust features, thus likewise yielding poor performance on $\mathcal{D}$.

The above two attacks are legal because the delusive distributions are close enough to the original distribution, that is, $\text{W}_\infty(\mathcal{D}, \widehat{\mathcal{D}}_1) \leq \epsilon$ and $\text{W}_\infty(\mathcal{D}, \widehat{\mathcal{D}}_2) \leq \epsilon$. Meanwhile, the attacks are also harmful. The following theorem directly compares the destructiveness of the attacks.

**Theorem 2.** *Let $f_{\mathcal{D}}$, $f_{\widehat{\mathcal{D}}_1}$, and $f_{\widehat{\mathcal{D}}_2}$ be the Bayes optimal classifiers for the mixture-Gaussian distributions $\mathcal{D}$, $\widehat{\mathcal{D}}_1$, and $\widehat{\mathcal{D}}_2$, defined in Eqs. (5), (6), and (7), respectively. For any $\eta > 0$, we have*

$$\mathcal{R}_{\text{nat}}(f_{\mathcal{D}}, \mathcal{D}) < \mathcal{R}_{\text{nat}}(f_{\widehat{\mathcal{D}}_2}, \mathcal{D}) < \mathcal{R}_{\text{nat}}(f_{\widehat{\mathcal{D}}_1}, \mathcal{D}).$$

The proof is provided in Appendix C.2. Theorem 2 indicates that both attacks will increase the natural risk of the Bayes optimal classifier. Moreover, $\widehat{\mathcal{D}}_1$ is more harmful because it always incurs higher natural risk than $\widehat{\mathcal{D}}_2$. The destructiveness depends on the dimension of non-robust features. For intuitive understanding, we plot the natural accuracy of the classifiers as a function of $d$ in Figure 2. We observe that, as the number of non-robust features increases, the natural accuracy of the standard model $f_{\widehat{\mathcal{D}}_1}$ continues to decline, while the natural accuracy of $f_{\widehat{\mathcal{D}}_2}$ first decreases and then increases.

**Adversarial training matters.** Adversarial training with proper $\epsilon$ will mitigate the reliance on non-robust features. For $\widehat{\mathcal{D}}_1$ the internal mechanism is similar to the case in Tsipras et al. [119], while for $\widehat{\mathcal{D}}_2$ the mechanism is different, and there was no such analysis before. Specifically, the optimal linear $\ell_\infty$ robust classifier (i.e., minimization of the adversarial risk with 0-1 loss) for $\widehat{\mathcal{D}}_1$ will rely solely on the robust feature. In contrast, the optimal robust classifier for $\widehat{\mathcal{D}}_2$ will rely on both robust and non-robust features, but the excessive reliance on non-robust features is mitigated. Hence, adversarial training matters in both cases and achieves better natural accuracy when compared with standard training. We make this formal in the following theorem.

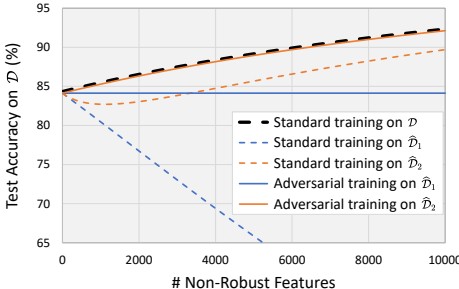

Figure 2: The natural accuracy of five models trained on the mixture-Gaussian distributions as a function of the number of non-robust features. As a concrete example, here we set $\sigma = 1$, $\eta = 0.01$ and vary $d$.

**Theorem 3.** *Let $f_{\widehat{\mathcal{D}}_1,\text{rob}}$ and $f_{\widehat{\mathcal{D}}_2,\text{rob}}$ be the optimal linear $\ell_\infty$ robust classifiers for the delusive distributions $\widehat{\mathcal{D}}_1$ and $\widehat{\mathcal{D}}_2$, defined in Eqs. (6) and (7), respectively. For any $0 < \eta < 1/3$, we have*

$$\mathcal{R}_{\text{nat}}(f_{\widehat{\mathcal{D}}_1}, \mathcal{D}) > \mathcal{R}_{\text{nat}}(f_{\widehat{\mathcal{D}}_1,\text{rob}}, \mathcal{D}) \quad \text{and} \quad \mathcal{R}_{\text{nat}}(f_{\widehat{\mathcal{D}}_2}, \mathcal{D}) > \mathcal{R}_{\text{nat}}(f_{\widehat{\mathcal{D}}_2,\text{rob}}, \mathcal{D}).$$

The proof is provided in Appendix C.3. Theorem 3 indicates that robust models achieve lower natural risk than standard models under delusive attacks. This is also reflected in Figure 2: After adversarial training on $\widehat{\mathcal{D}}_1$, natural accuracy is largely recovered and keeps unchanged as $d$ increases. While on $\widehat{\mathcal{D}}_2$, natural accuracy can be recovered better and keeps increasing as $d$ increases. Beyond the theoretical analyses for these simple cases, we also observe that the phenomena in Theorem 2 and Theorem 3 generalize well to our empirical experiments on real-world datasets in Section 5.1.

## 4 Practical Attacks for Testing Defense

Here we briefly describe five heuristic attacks. A detailed description is deferred to Appendix D. The five attacks along with the L2C attack proposed by Feng et al. [32] will be used in next section for validating the destructiveness of delusive attacks and thus the necessity of adversarial training.

In practice, we focus on the empirical distribution $\mathcal{D}_n$ over $n$ data points drawn from $\mathcal{D}$. Inspired by "non-robust features suffice for classification" [56], we propose to construct delusive perturbations by injecting non-robust features correlated consistently with a specific label. Given a standard model $f_{\mathcal{D}}$ trained on $\mathcal{D}_n$, the attacks perturb each input $\boldsymbol{x}$ (with label $y$) in $\mathcal{D}_n$ as follows:

- P1: **Adversarial perturbations.** It adds a small adversarial perturbation to $\boldsymbol{x}$ to ensure that it is misclassified as a target $t$ by minimizing $\ell\left(f_{\mathcal{D}}(\boldsymbol{x}'), t\right)$ such that $\boldsymbol{x}' \in \mathcal{B}(\boldsymbol{x}, \epsilon)$, where $t$ is chosen deterministially based on $y$.

- P2: **Hypocritical perturbations.** It adds a small helpful perturbation to $\boldsymbol{x}$ by minimizing $\ell\left(f_{\mathcal{D}}(\boldsymbol{x}'), y\right)$ such that $\boldsymbol{x}' \in \mathcal{B}(\boldsymbol{x}, \epsilon)$, so that the standard model could easily produce a correct prediction.

- P3: **Universal adversarial perturbations.** This attack is a variant of P1. It adds the class-specific universal adversarial perturbation $\boldsymbol{\xi}_t$ to $\boldsymbol{x}$. All inputs with the same label $y$ are perturbed with the same perturbation $\boldsymbol{\xi}_t$, where $t$ is chosen deterministially based on $y$.

- P4: **Universal hypocritical perturbations.** This attack is a variant of P2. It adds the class-specific universal helpful perturbation $\boldsymbol{\xi}_y$ to $\boldsymbol{x}$. All inputs with the same label $y$ are perturbed with the same perturbation $\boldsymbol{\xi}_y$.

- P5: **Universal random perturbations.** This attack injects class-specific random perturbation $\boldsymbol{r}_y$ to each $\boldsymbol{x}$. All inputs with the label $y$ is perturbed with the same perturbation $\boldsymbol{r}_y$. Despite the simplicity of this attack, we find that it are surprisingly effective in some cases.

Figure 3 visualizes the universal perturbations for different datasets and threat models. The perturbed inputs and their original labels constitute the perturbed datasets $\widehat{\mathcal{D}}_{\mathsf{P1}} \sim \widehat{\mathcal{D}}_{\mathsf{P5}}$.

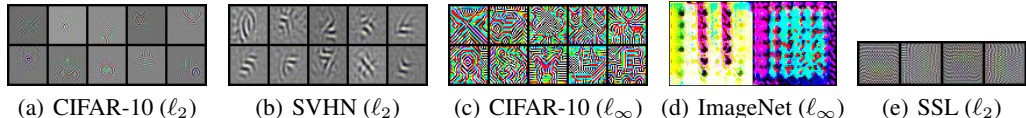

(a) CIFAR-10 ($\ell_2$)   (b) SVHN ($\ell_2$)   (c) CIFAR-10 ($\ell_\infty$)   (d) ImageNet ($\ell_\infty$)   (e) SSL ($\ell_2$)

Figure 3: Universal perturbations for the P3 and P4 attacks across different datasets and threat models. Perturbations are rescaled to lie in the $[0, 1]$ range for display. The resulting inputs are nearly indistinguishable from the originals to a human observer (see Appendix B Figures 10, 11, and 12).

## 5 Experiments

In order to demonstrate the effectiveness and versatility of the proposed defense, we conduct experiments on CIFAR-10 [63], SVHN [81], a subset of ImageNet [95], and MNIST-CIFAR [104] datasets. More details on experimental setup are provided in Appendix A. Our code is available at https://github.com/TLMichael/Delusive-Adversary.

Firstly, we perform a set of experiments on supervised learning to provide a comprehensive understanding of delusive attacks (Section 5.1). Secondly, we demonstrate that the delusive attacks can also obstruct rotation-based self-supervised learning (SSL) [41] and adversarial training also helps a lot in this case (Section 5.2). Finally, we show that adversarial training is a promising method to overcome the simplicity bias on the MNIST-CIFAR dataset [104] if the $\epsilon$-ball is chosen properly (Section 5.3).

## 5.1 Understanding Delusive Attacks

Here, we investigate delusive attacks from six different perspectives: *i)* baseline results on CIFAR-10, *ii)* transferability of delusive perturbations to various architectures, *iii)* performance changes of various defender's budgets, *iv)* a simple countermeasure, *v)* comparison with Feng et al. [32], and *vi)* performance of other adversarial training variants.

**Baseline results.** We consider the typical $\ell_2$ threat model with $\epsilon = 0.5$ for CIFAR-10 by following [56]. We use the attacks described in Section 4 to generate the delusive perturbations. To execute the attacks P1 $\sim$ P4, we pre-train a VGG-16 [108] as the standard model $f_\mathcal{D}$ using standard training on the original training set. We then perform standard training and adversarial training on the delusive datasets $\widehat{\mathcal{D}}_{\text{P1}} \sim \widehat{\mathcal{D}}_{\text{P5}}$. Standard data augmentation (i.e., cropping, mirroring) is adopted. The natural accuracy of the models is evaluated on the clean test set of CIFAR-10.

Results are summarized in Figure 4. We observe that the natural accuracy of the standard models dramatically decreases when training on the delusive datasets, especially on $\widehat{\mathcal{D}}_{\text{P3}}$, $\widehat{\mathcal{D}}_{\text{P4}}$ and $\widehat{\mathcal{D}}_{\text{P5}}$. The most striking observation to emerge from the results is the effectiveness of the P5 attack. It seems that the naturally trained model seems to rely exclusively on the small random patterns in this case, even though there are still abundant natural features in $\widehat{\mathcal{D}}_{\text{P5}}$. Such behaviors resemble the conjunction learner[1] studied in the pioneering work [82], where they showed that such a learner

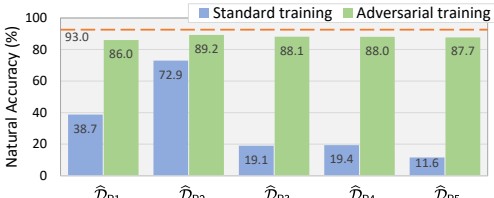

Figure 4: Natural accuracy on CIFAR-10 using VGG-16 under $\ell_2$ threat model. The horizontal line indicates the natural accuracy of a standard model trained on the clean training set.

is highly vulnerable to delusive attacks. Also, we point out that such behaviors could be attributed to the gradient starvation [91] and simplicity bias [104] phenomena of neural networks. These recent studies both show that neural networks trained by SGD preferentially capture a subset of features relevant for the task, despite the presence of other predictive features that fail to be discovered [50].

Anyway, our results demonstrate that adversarial training can successfully eliminate the delusive features within the $\epsilon$-ball. As shown in Figure 4, natural accuracy can be significantly improved by adversarial training in all cases. Besides, we observe that P1 is more destructive than P2, which is consistent with our theoretical analysis of the hypothetical settings in Section 3.2.

**Evaluation of transferability.** A more realistic setting is to attack different classifiers using the same delusive perturbations. We consider various architectures including VGG-19, ResNet-18, ResNet-50, and DenseNet-121 as victim classifiers. The delusive datasets are the same as in the baseline experiments. Results are deferred to Figure 8 in Appendix B. We observe that the attacks have good transferability across the architectures, and again, adversarial training can substantially improve natural accuracy in all cases. One exception is that the P5 attack is invalid for DenseNet-121. A possible explanation for this might be that the simplicity bias of DenseNet-121 on random patterns is minor. This means that different architectures may have distinct simplicity biases. Due to space constraints, a detailed investigation is out of the scope of this work.

**What if the threat model is not specified?** Our theoretical analysis in Section 3.1 highlights the importance of choosing a proper budget $\epsilon$ for AT. Here, we try to explore this situation where the threat model is not specified by varying the defender's budget. Results on CIFAR-10 using ResNet-18 under $\ell_2$ threat model are summarized in Figure 5. We observe that a budget that is too large may slightly hurt performance, while a budget that is too small is not enough to mitigate the attacks. Empirically, the optimal budget for P3 and P4 is about $0.4$ and for P1 and P2 it is about $0.25$. P5 is the easiest to defend—AT with a small budget (about $0.1$) can significantly mitigate its effect.

**A simple countermeasure.** In addition to adversarial training, a simple countermeasure is adding clean data to the training set. This will neutralize the perturbed data and make it closer to the original distribution. We explore this countermeasure on SVHN since extensive extra training examples are available in that dataset. Results are summarized in Figure 6. We observe that the performance of standard training is improved with the increase of the number of additional clean examples, and the

---

[1]The conjunction learner first identifies a subset of features that appears in every examples of a class, then classifies an example as the class if and only if it contains such features [82].

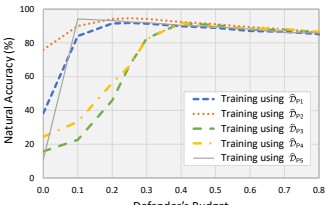
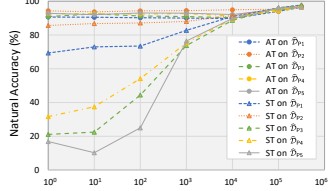
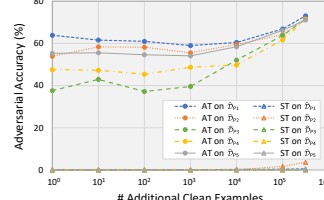

Figure 5: Natural accuracy as a function of the defender's budget on CIFAR-10.

Figure 6: Natural accuracy (left) and adversarial accuracy (right) as a function of the number of additional clean examples on SVHN using ResNet-18 under $\ell_2$ threat model.

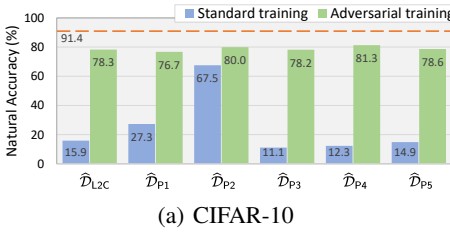

(a) CIFAR-10

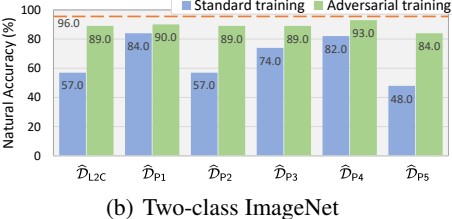

(b) Two-class ImageNet

Figure 7: Natural accuracy on CIFAR-10 using VGG-11 (left) and two-class ImageNet using ResNet-18 (right) under $\ell_\infty$ threat model. The horizontal orange line indicates the natural accuracy of a standard model trained on the clean training set.

performance of adversarial training can also be improved with more data. Overall, it is recommend that combining this simple countermeasure with adversarial training to further improve natural accuracy. Besides the focus on natural accuracy in this work, another interesting measure is the model accuracy on adversarial examples. It turns out that adversarial accuracy of the models can also be improved with more data. We also observe that different delusive attacks have different effects on the adversarial accuracy. A further study with more focus on adversarial accuracy is therefore suggested.

**Comparison with L2C.** We compare the heuristic attacks with the L2C attack proposed by Feng et al. [32] and, show that adversarial training can mitigate all these attacks. Following their settings on CIFAR-10 and a two-class ImageNet, the $\ell_\infty$-norm bounded threat models with $\epsilon = 0.032$ and $\epsilon = 0.1$ are considered. The victim classifier is VGG-11 and ResNet-18 for CIFAR-10 and the two-class ImageNet, respectively. Table 1 reports the time cost for executing six attack methods on CIFAR-10. We find that the heuristic attacks are significantly faster than L2C, since the bi-level optimization process in L2C is extremely time-consuming. Figure 7 shows the performance of stan-

Table 1: Comparison of time cost. The L2C attack needs to train an autoencoder to generate perturbations. The P1 $\sim$ P4 attacks need to train a standard classifier to generate perturbations, and P5 needs not.

| Method | Time Cost (min) | | |
|---|---|---|---|
| | Training | Generating | Total |
| L2C | 7411.5 | 0.4 | 7411.9 |
| P1 / P2 | 25.9 | 12.6 | 38.5 |
| P3 / P4 | 25.9 | 4.6 | 30.5 |
| P5 | 0.0 | 0.1 | 0.1 |

dard training and adversarial training on delusive datasets. The results indicate that most of the heuristic attacks are comparable with L2C, and AT can improve natural accuracy in all cases.

**Performance of adversarial training variants.** It is noteworthy that AT variants are also effective in our setting, since they aim to tackle the adversarial risk. To support this, we consider instance-dependent-based variants (such as MART [125], GAIRAT [141], and MAIL [123]) and curriculum-based variants (such as CAT [13], DAT [124], FAT [140]). Specifically, we chose to experiment with the currently most effective variants among them (i.e., GAIRAT and FAT, according to the latest leaderboard at RobustBench [21]). Additionally, we consider random noise training (denoted as RandNoise) using the uniform noise within the $\epsilon$-ball for comparison. We also report the results of standard training (denoted as ST) and the conventional PGD-based AT [74] (denoted as PGD-AT) for reference. The results are summarized in Table 2. We observe that the performance of random noise training is marginal. In contrast, all AT methods show significant improvements, thanks to the theoretical analysis provided by Theorem 1. Besides, we observe that FAT achieves overall better results than other AT variants. This may be due to the tight upper bound of the adversarial risk pursued by FAT. In summary, these results successfully validate the effectiveness of AT variants.

Table 2: Natural accuracy on CIFAR-10 using ResNet-18 under $\ell_\infty$ threat model with $\epsilon = 8/255$. The column of "`Clean`" denotes the natural accuracy of the models trained on the clean training set.

| Method | Clean | L2C | P1 | P2 | P3 | P4 | P5 |
|---|---|---|---|---|---|---|---|
| ST | **94.62** | 15.76 | 15.70 | 61.35 | 9.40 | 13.58 | 10.12 |
| RandNoise | 94.26 | 17.10 | 17.32 | 63.36 | 10.52 | 14.37 | 27.56 |
| PGD-AT | 85.18 | 82.84 | 84.18 | 86.74 | **86.37** | 83.18 | 84.57 |
| GAIRAT | 81.90 | 79.96 | 79.61 | 82.68 | 82.05 | 82.81 | 82.28 |
| FAT | 87.43 | **85.51** | **86.05** | **88.98** | 84.39 | **84.22** | **87.78** |

Table 3: Adversarial training on MNIST-CIFAR: The table reports test accuracy on the MNIST-CIFAR test set and the MNIST-randomized test set. Our customized AT successfully overcomes SB, while others not. The MNIST-randomized accuracy indicates that our adversarially trained models achieve nontrivial performance when there are only CIFAR features exist in the inputs.

| Model | Test Accuracy on MNIST-CIFAR | | | MNIST-Randomized Accuracy | | |
|---|---|---|---|---|---|---|
| | ST | AT [104] | AT (ours) | ST | AT [104] | AT (ours) |
| VGG-16 | 99.9 | 100.0 | 91.3 | 49.1 | 51.6 | **91.2** |
| ResNet-50 | 100.0 | 99.9 | 89.7 | 48.9 | 49.2 | **88.6** |
| DenseNet-121 | 100.0 | 100.0 | 91.5 | 48.8 | 49.2 | **90.8** |

## 5.2 Evaluation on Rotation-based Self-supervised Learning

To further show the versatility of the attacks and defense, we conduct experiments on rotation-based self-supervised learning (SSL) [41], a process that learns representations by predicting rotation angles (0°, 90°, 180°, 270°). SSL methods seem to be inherently resist to the poisoning attacks that require mislabeling, since they do not use human-annotated labels to learn representations. Here, we examine whether SSL can resist the delusive attacks. We use delusive attacks to perturb the training data for the pretext task. To evaluate the quality of the learned representations, the downstream task is trained on the clean data using logistic regression. Results are deferred to Figure 9 in Appendix B.

We observe that the learning of the pretext task can be largely hijacked by the attacks. Thus the learned representations are poor for the downstream task. Again, adversarial training can significantly improve natural accuracy in all cases. An interesting observation from Figure 9(b) is that the quality of the adversarially learned representations is slightly better than that of standard models trained on the original training set. This is consistent with recent hypotheses stating that robust models may transfer better [98, 121, 69, 22, 2]. These results show the possibility of delusive attacks and defenses for SSL, and suggest that studying the robustness of other SSL methods [58, 17] against data poisoning is a promising direction for future research.

## 5.3 Overcoming Simplicity Bias

A recent work by Shah et al. [104] proposed the MNIST-CIFAR dataset to demonstrate the simplicity bias (SB) of using standard training to learn neural networks. Specifically, the MNIST-CIFAR images $x$ are vertically concatenations of the "simple" MNIST images $x_m$ and the more complex CIFAR-10 images $x_c$ (i.e., $x = [x_m; x_c]$). They found that standard models trained on MNIST-CIFAR will exclusively rely on the MNIST features and remain invariant to the CIFAR features. Thus randomizing the MNIST features drops the model accuracy to random guessing.

From the perspective of delusive adversaries, we can regard the MNIST-CIFAR dataset as a delusive version of the original CIFAR dataset. Thus, AT should mitigate the delusive perturbations, as Theorem 1 pointed out. However, Shah et al. [104] tried AT on MNIST-CIFAR yet failed. Contrary to their results, here we demonstrate that AT is actually workable. The key factor is the choice of the threat model. They failed because they chose an improper ball $\mathcal{B}(x, \epsilon) = \{x' \in \mathcal{X} : \|x - x'\|_\infty \leq 0.3\}$, while we set $\mathcal{B}(x, \epsilon) = \{x' \in \mathcal{X} : \|x_m - x'_m\|_\infty + \infty \cdot \|x_c - x'_c\|_\infty \leq 1\}$. Our choice forces the space of MNIST features to be a non-robust region during AT, and prohibits the CIFAR features from being perturbed. Results are summarized in Table 3. We observe that our choice leads to models that do not rely on the simple MNIST features, thus AT can eliminate the simplicity bias.

# 6 Related Work

**Data poisoning.** The main focus of this paper is the threat of delusive attacks [82], which belongs to data poisoning attacks [6, 7, 43]. Generally speaking, data poisoning attacks manipulate the training data to cause a model to fail during inference. Both *integrity* and *availability* attacks were extensively studied for classical models [5, 42, 82, 80, 8, 9, 76, 132, 143]. For neural networks, most of the existing works focused on targeted misclassification [61, 103, 145, 1, 55, 39, 105, 20, 92] and *backdoor* attacks [47, 18, 71, 120, 72, 83, 86, 84], while there was little work on availability attacks [102, 78, 32, 106]. Recently, Feng et al. [32] showed that availability attacks are feasible for deep networks. This paper follows their setting where the perturbed training data is correctly labeled. We further point out that their studied threat is exactly the *delusive adversary* (a.k.a. clean-label availability attacks), which was previously considered for classical models [82]. Besides, other novel directions of data poisoning are rapidly evolving such as semi-supervised learning [70, 35, 14], contrastive learning [15, 96], domain adaptation [75], and online learning [88], etc.

**Existing defenses.** There were many defense strategies proposed for defending against targeted attacks and backdoor attacks, including detection-based defenses [111, 118, 16, 23, 38, 90, 48, 28], randomized smoothing [94, 127], differential privacy [73, 51], robust training [12, 66, 67, 40], and model repairing [19, 68, 130], while some of them may be overwhelmed by adaptive attacks [62, 122, 107]. Robust learnability under data poisoning attacks can be analyzed from theoretical aspects [11, 126, 36]. Similarly, our proposed defense is principled and theoretically justified. More importantly, previous defenses mostly focus on defending against integrity attacks, while none of them are specially designed to resist delusive attacks. The work most similar to ours is that of Farokhi [31]. They only handle the *linear regression* model by relaxing distributionally robust optimization (DRO) as a regularizer, while we can tackle delusive attacks for *any* classifier.

**Adversarial training.** Since the discovery of adversarial examples (a.k.a. evasion attacks at test time) in neural networks [10, 113], plenty of defense methods have been proposed to mitigate this vulnerability [44, 89, 3]. Among them, adversarial training is practically considered as a principled defense against test-time adversarial examples [74, 21] at the price of slightly worse natural accuracy [112, 119, 135] and moderate robust generalization [101, 93], and many variants were devoted to improving the performance [139, 140, 26, 131, 87, 37, 29, 114, 54, 45, 27]. Besides, it has been found that adversarial training may intensify backdoor attacks in experiments [128]. In contrast, both of our theoretical and empirical evidences suggest that adversarial training can mitigate delusive attacks. On the other hand, adversarial training also led to further benefits in robustness to noisy labels [146], out-of-distribution generalization [133, 136, 60], transfer learning [98, 116, 121], domain adaption [4], novelty detection [46, 97], explainability [142, 85], and image synthesis [100].

**Concurrent work.** The threat of delusive attacks [82, 32] is attracting attention from the community. Several attack techniques are concurrently and independently proposed, including Unlearnable Examples [53], Alignment [33], NTGA [137], Adversarial Shortcuts [30], and Adversarial Poisoning [34]. Contrary to them, we mainly focus on introducing a principled defense method (i.e., adversarial training), while as by-products, five delusive attacks are presented and investigated in this paper. Meanwhile, Huang et al. [53] and Fowl et al. [34] also experiment with adversarial training, but *only* for their proposed delusive attacks. In contrast, this paper not only provides empirical evidence on the success of adversarial training against *six* different delusive attacks, but also offers theoretical justifications for the defense, which is of great significance to security. We believe that our findings will promote the use of adversarial training in practical applications in the future.

# 7 Conclusion and Future Work

In this paper, we suggest applying adversarial training in practical applications, rather than standard training whose performance risks being substantially deteriorated by delusive attacks. Both theoretical and empirical results vote for adversarial training when confronted with delusive adversaries. Nonetheless, some limitations remain and may lead to future directions: *i)* Our implementation of adversarial training adopts the popular PGD-AT framework [74, 87], which could be replaced with certified training methods [129, 138] for better robustness guarantee. *ii)* The success of our proposed defense relies on generalization, just like most ML algorithms, so an analysis of robust generalization error bound for this case would be useful. *iii)* Adversarial training may increase the disparity of accuracy between groups [134, 117], which could be mitigated by fair robust learning [134].

## Acknowledgments and Disclosure of Funding

This work was supported by the National Natural Science Foundation of China (Grant No. 62076124, 62076128) and the National Key R&D Program of China (2020AAA0107000).

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
