# Supplementary Material:

# Better Safe Than Sorry: Preventing Delusive Adversaries with Adversarial Training

## A    Experimental Setup

Experiments run with NVIDIA GeForce RTX 2080 Ti GPUs. We report the number with a single run of experiments. Our implementation is based on PyTorch, and the code to reproduce our results is available at https://github.com/TLMichael/Delusive-Adversary.

### A.1    Datasets and Models

Table 4 reports the parameters used for the datasets and models.

**CIFAR-10[2].**    This dataset [63] consists of 60,000 $32 \times 32$ colour images (50,000 images for training and 10,000 images for testing) in 10 classes ("airplane", "car", "bird", "cat", "deer", "dog", "frog", "horse", "ship", and "truck"). Early stopping is done with holding out 1000 examples from the training set. We use various architectures for this dataset, including VGG-11, VGG-16, VGG-19 [108], ResNet-18, ResNet-50 [49], and DenseNet-121 [52]. The initial learning rate is set to 0.1. For supervised learning on this dataset, we run 150 epochs on the training set, where we decay the learning rate by a factor 0.1 in the 100th and 125th epochs. For self-supervised learning, we run 70 epochs, where we decay the learning rate by a factor 0.1 in the 40th and 55th epochs. The license for this dataset is unknown[3].

**SVHN[4].**    This dataset [81] consists of 630,420 $32 \times 32$ colour images (73,257 images for training, 26,032 images for testing, and 531,131 images to use as extra training data) in 10 classes ("0", "1", "2", "3", "4", "5", "6", "7", "8", and "9"). Early stopping is done with holding out 1000 examples from the training set. We use the ResNet-18 architecture for this dataset. The initial learning rate is set to 0.01. We run 50 epochs on the training set, where we decay the learning rate by a factor 0.1 in the 30th and 40th epochs. The license for this dataset is custom (non-commercial)[5].

**Two-class ImageNet[6].**    Following Feng et al. [32], this dataset is a subset of ImageNet [95] that consists of 2,700 $224 \times 224$ colour images (2,600 images for training and 100 images for testing) in 2 classes ("bulbul", "jellyfish"). Early stopping is done with holding out 380 examples from the training set. We use the ResNet-18 architecture for this dataset. The initial learning rate is set to 0.1. We run 100 epochs on the training set, where we decay the learning rate by a factor 0.1 in the 75th and 90th epochs. The license for this dataset is custom (research, non-commercial)[7].

**MNIST-CIFAR[8].**    Following Shah et al. [104], this dataset consists of 11,960 $64 \times 32$ colour images (10,000 images for training and 1,960 images for testing) in 2 classes: images in class $-1$ and class 1 are vertical concatenations of MNIST digit zero & CIFAR-10 car and MNIST digit one & CIFAR-10 truck images, respectively. We use various architectures for this dataset, including VGG-16, ResNet-50, and DenseNet-121. The initial learning rate is set to 0.05. We run 100 epochs on the training set, where we decay the learning rate by a factor 0.2 in the 50th and 150th epochs and a factor 0.5 in the 100th epoch. The license for this dataset is unknown[9].

---

[2] https://www.cs.toronto.edu/~kriz/cifar.html
[3] https://paperswithcode.com/dataset/cifar-10
[4] http://ufldl.stanford.edu/housenumbers/
[5] https://paperswithcode.com/dataset/svhn
[6] https://github.com/kingfengji/DeepConfuse
[7] https://paperswithcode.com/dataset/imagenet
[8] https://github.com/harshays/simplicitybiaspitfalls
[9] https://paperswithcode.com/dataset/mnist

| Parameter | CIFAR-10 | SVHN | Two-class ImageNet | MNIST-CIFAR |
|---|---|---|---|---|
| # training examples | 50,000 | 73,257 | 2,600 | 10,000 |
| # test examples | 10,000 | 26,032 | 100 | 1,960 |
| # features | 3,072 | 3,072 | 150,528 | 6,144 |
| # classes | 10 | 10 | 2 | 2 |
| batch size | 128 | 128 | 32 | 256 |
| learning rate | 0.1 | 0.01 | 0.1 | 0.05 |
| SGD momentum | 0.9 | 0.9 | 0.9 | 0.9 |
| weight decay | $5 \cdot 10^{-4}$ | $5 \cdot 10^{-4}$ | $5 \cdot 10^{-4}$ | $5 \cdot 10^{-5}$ |

Table 4: Experimental setup and parameters for the each dataset.

## A.2 Adversarial Training

Unless otherwise specified, we perform adversarial training to train robust classifiers by following Madry et al. [74]. Specifically, we train against a projected gradient descent (PGD) adversary, starting from a random initial perturbation of the training data. We consider adversarial perturbations in $\ell_p$ norm where $p = \{2, \infty\}$. Unless otherwise specified, we use the values of $\epsilon$ provided in Table 5 to train our models. We use 7 steps of PGD with a step size of $\epsilon/5$. For overcoming simplicity bias on MNIST-CIFAR, we modify the original $\epsilon$-ball used in Shah et al. [104] (i.e., $\mathcal{B}(\boldsymbol{x}, \epsilon) = \{\boldsymbol{x}' \in \mathcal{X} : \|\boldsymbol{x} - \boldsymbol{x}'\|_\infty \leq 0.3\}$) to the entire space of the MNIST features $\mathcal{B}(\boldsymbol{x}, \epsilon) = \{\boldsymbol{x}' \in \mathcal{X} : \|\boldsymbol{x}_m - \boldsymbol{x}'_m\|_\infty + \infty \cdot \|\boldsymbol{x}_c - \boldsymbol{x}'_c\|_\infty \leq 1\}$, where $\boldsymbol{x}$ represents the vertical concatenation of a MNIST image $\boldsymbol{x}_m$ and a CIFAR-10 image $\boldsymbol{x}_c$.

| Adversary | CIFAR-10 | SVHN | Two-class ImageNet |
|---|---|---|---|
| $\ell_\infty$ | 0.032 | - | 0.1 |
| $\ell_2$ | 0.5 | 0.5 | - |

Table 5: Value of $\epsilon$ used for adversarial training of each dataset and $\ell_p$ norm.

## A.3 Delusive Adversaries

Six delusive attacks are considered to validate our proposed defense. We reimplement the L2C attack [32] using the code provided by the authors[10]. The other five attacks are constructed as follows. To execute P1 $\sim$ P4, we perform normalized gradient descent ($\ell_p$-norm of gradient is fixed to be constant at each step). At each step we clip the input to in the $[0, 1]$ range so as to ensure that it is a valid image. To execute P5, noises are sampled from Gaussian distribution and then projected to the ball for $\ell_2$-norm bounded perturbations; for $\ell_\infty$-norm bounded perturbations, noises are directly sampled from a uniform distribution. Unless otherwise specified, the attacker's $\epsilon$ are the same with adversarial training used by the defender. Details on the optimization procedure are shown in Table 6.

| Parameter | P1 | P2 | P3 | P4 | P5 |
|---|---|---|---|---|---|
| step size | $\epsilon/5$ | $\epsilon/5$ | $\epsilon/5$ | $\epsilon/5$ | $\epsilon$ |
| iterations | 100 | 100 | 500 | 500 | 1 |

Table 6: Parameters used for optimization procedure to construct each delusive dataset in Section 4.

# B  Omitted Figures

---

[10]https://github.com/kingfengji/DeepConfuse

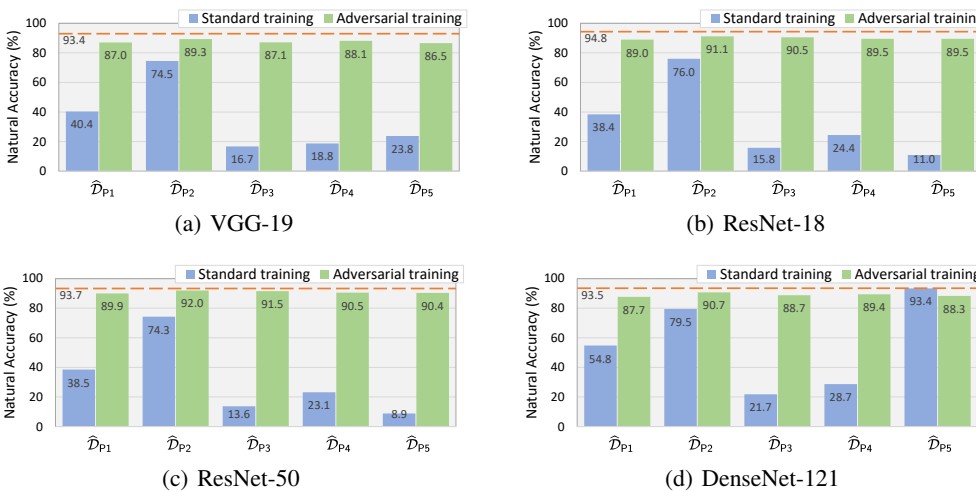

Figure 8: Natural accuracy on CIFAR-10 under $\ell_2$ threat model. The horizontal orange line indicates the natural accuracy of a standard model trained on the clean training set.

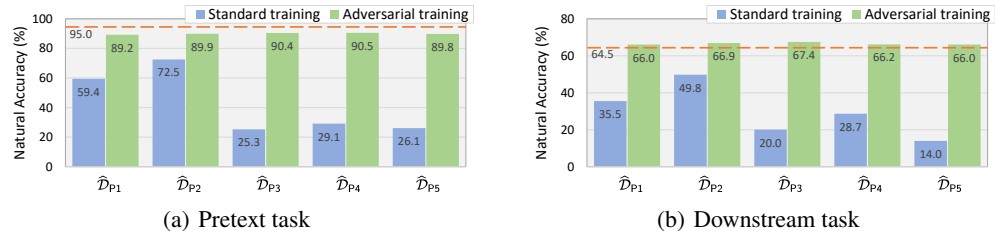

Figure 9: Rotation-based SSL on CIFAR-10 using ResNet-18 under $\ell_2$ threat model. The horizontal line indicates the natural accuracy of a standard model trained on the clean training set.

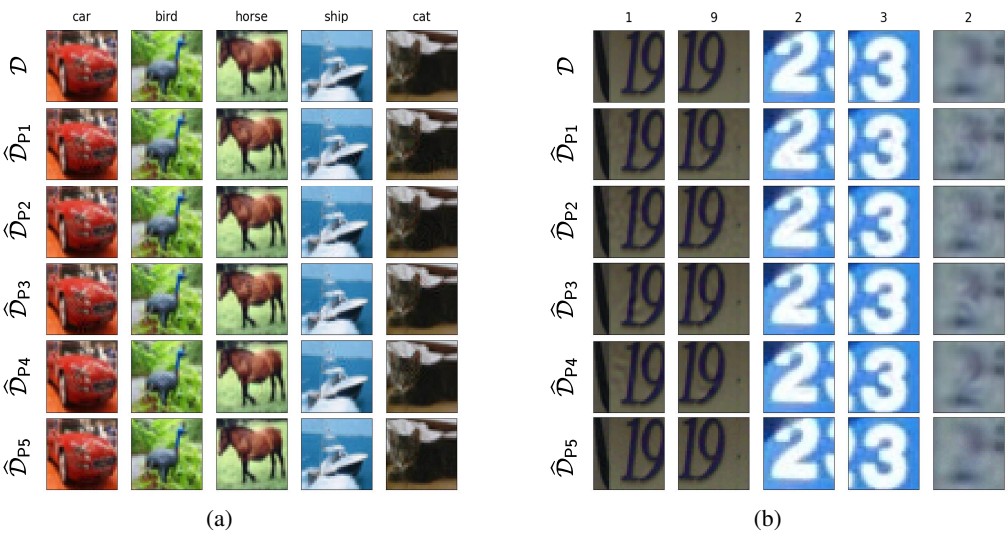

Figure 10: **Left**: Random samples from the CIFAR-10 training set: the original training $\mathcal{D}$; the perturbed training sets $\widehat{\mathcal{D}}_{P1}$, $\widehat{\mathcal{D}}_{P2}$, $\widehat{\mathcal{D}}_{P3}$, $\widehat{\mathcal{D}}_{P4}$, and $\widehat{\mathcal{D}}_{P5}$. The threat model is the $\ell_2$ ball with $\epsilon = 0.5$. **Right**: First five examples from the SVHN training set: the original training $\mathcal{D}$; the perturbed training sets $\widehat{\mathcal{D}}_{P1}$, $\widehat{\mathcal{D}}_{P2}$, $\widehat{\mathcal{D}}_{P3}$, $\widehat{\mathcal{D}}_{P4}$, and $\widehat{\mathcal{D}}_{P5}$. The threat model is the $\ell_2$ ball with $\epsilon = 0.5$.

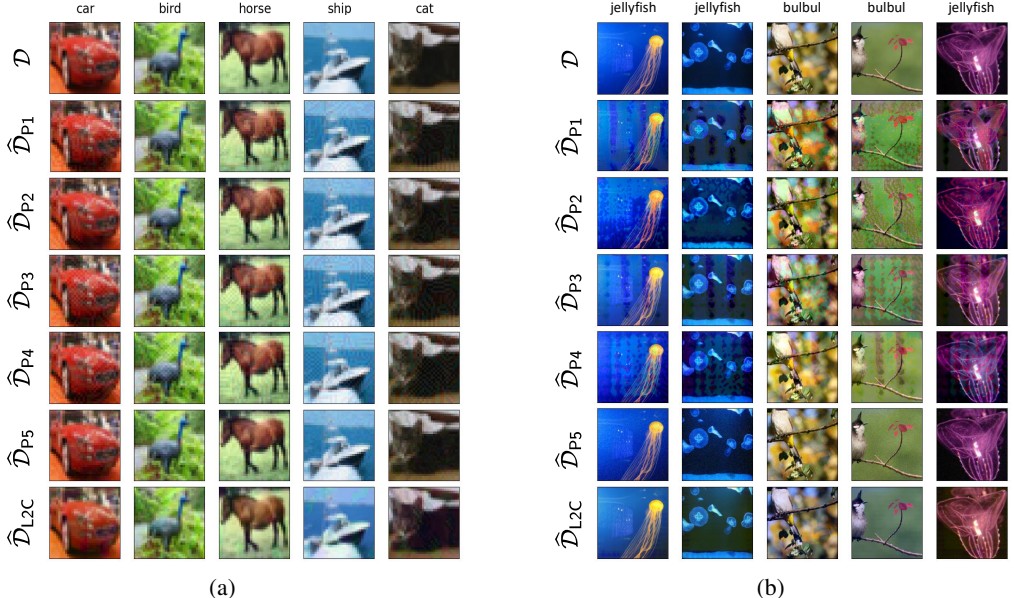

(a)            (b)

Figure 11: **Left**: Random samples from the CIFAR-10 training set: the original training $\mathcal{D}$; the perturbed training sets $\widehat{\mathcal{D}}_{\mathsf{P1}}$, $\widehat{\mathcal{D}}_{\mathsf{P2}}$, $\widehat{\mathcal{D}}_{\mathsf{P3}}$, $\widehat{\mathcal{D}}_{\mathsf{P4}}$, and $\widehat{\mathcal{D}}_{\mathsf{P5}}$. The threat model is the $\ell_\infty$ ball with $\epsilon = 0.032$. **Right**: First five examples from the two-class ImageNet training set: the original training $\mathcal{D}$; the perturbed training sets $\widehat{\mathcal{D}}_{\mathsf{P1}}$, $\widehat{\mathcal{D}}_{\mathsf{P2}}$, $\widehat{\mathcal{D}}_{\mathsf{P3}}$, $\widehat{\mathcal{D}}_{\mathsf{P4}}$, and $\widehat{\mathcal{D}}_{\mathsf{P5}}$. The threat model is the $\ell_\infty$ ball with $\epsilon = 0.1$.

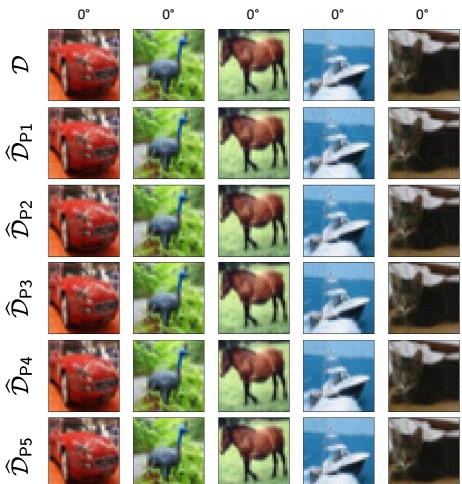

Figure 12: Random samples from the CIFAR-10 training set for rotation-based self-supervised learning: the original training $\mathcal{D}$; the perturbed training sets $\widehat{\mathcal{D}}_{\mathsf{P1}}$, $\widehat{\mathcal{D}}_{\mathsf{P2}}$, $\widehat{\mathcal{D}}_{\mathsf{P3}}$, $\widehat{\mathcal{D}}_{\mathsf{P4}}$, and $\widehat{\mathcal{D}}_{\mathsf{P5}}$. The threat model is the $\ell_2$ ball with $\epsilon = 0.5$.

# C Proofs

In this section, we provide the proofs of our theoretical results in Section 3.

## C.1 Proof of Theorem 1

The main tool in proving our key results is the following lemma, which characterizes the equivalence of adversarial risk and the DRO problem bounded in an $\infty$-Wasserstein ball.

**Lemma 4.** *Given a classifier $f : \mathcal{X} \to \mathcal{Y}$, for any data distribution $\mathcal{D}$, we have*

$$\mathbb{E}_{(\boldsymbol{x},y)\sim\mathcal{D}} \left[ \max_{\boldsymbol{x}'\in\mathcal{B}_{\epsilon}(\boldsymbol{x},\epsilon)} \ell(f(\boldsymbol{x}'),y) \right] = \max_{\mathcal{D}'\in\mathcal{B}_{\mathrm{W}_{\infty}}(\widehat{\mathcal{D}},\epsilon)} \mathbb{E}_{(\boldsymbol{x},y)\sim\mathcal{D}} [\ell(f(\boldsymbol{x}),y)]$$

This lemma is proved by Proposition 3.1 in Staib and Jegelka [110] and Lemma 3.3 in Zhu et al. [147], which indicates that adversarial training is actually equivalent to the DRO problem that minimizes the worst-case distribution constrained by the $\infty$-Wasserstein distance.

Theorem 1, restated below, shows that adversarial training on the poison data is optimizing an upper bound of natural risk on the original data.

**Theorem 1 (restated).** *Given a classifier $f : \mathcal{X} \to \mathcal{Y}$, for any data distribution $\mathcal{D}$ and any delusive distribution $\widehat{\mathcal{D}}$ such that $\widehat{\mathcal{D}} \in \mathcal{B}_{\mathrm{W}_{\infty}}(\mathcal{D}, \epsilon)$, we have*

$$\mathcal{R}_{\mathsf{nat}}(f,\mathcal{D}) \leq \max_{\mathcal{D}'\in\mathcal{B}_{\mathrm{W}_{\infty}}(\widehat{\mathcal{D}},\epsilon)} \mathcal{R}_{\mathsf{nat}}(f,\mathcal{D}') = \mathcal{R}_{\mathsf{adv}}(f,\widehat{\mathcal{D}}).$$

*Proof.* The first inequality comes from the symmetry of Wasserstein distance:

$$\mathrm{W}_{\infty}(\mathcal{D},\widehat{\mathcal{D}}) = \mathrm{W}_{\infty}(\widehat{\mathcal{D}},\mathcal{D}),$$

which means that the original distribution exists in the neighborhood of $\widehat{\mathcal{D}}$:

$$\mathcal{D} \in \mathcal{B}_{\mathrm{W}_{\infty}}(\widehat{\mathcal{D}},\epsilon).$$

Thus the natural risk on the original distribution can be upper bounded by DRO on the delusive distribution.

For the last equality, we simply use the fact in Lemma 4 that adversarial risk is equivalent to DRO defined with respect to the $\infty$-Wasserstein distance. This concludes the proof. $\square$

## C.2 Proof of Theorem 2

We first review the original mixture-Gaussian distribution $\mathcal{D}$, the corresponding delusive distribution $\widehat{\mathcal{D}}_1$, and $\widehat{\mathcal{D}}_2$.

The original mixture-Gaussian distribution $\mathcal{D}$:

$$y \overset{u\cdot a\cdot r}{\sim} \{-1,+1\}, \quad \boldsymbol{x} \sim \mathcal{N}(y\cdot\boldsymbol{\mu},\sigma^2\boldsymbol{I}), \quad \text{where } \boldsymbol{\mu} = (1,\eta,\ldots,\eta) \in \mathbb{R}^{d+1}. \tag{8}$$

The first delusive mixture-Gaussian distribution $\widehat{\mathcal{D}}_1$:

$$y \overset{u\cdot a\cdot r}{\sim} \{-1,+1\}, \quad \boldsymbol{x} \sim \mathcal{N}(y\cdot\widehat{\boldsymbol{\mu}}_1,\sigma^2\boldsymbol{I}), \quad \text{where } \widehat{\boldsymbol{\mu}}_1 = (1,-\eta,\ldots,-\eta) \in \mathbb{R}^{d+1}. \tag{9}$$

The second delusive mixture-Gaussian distribution $\widehat{\mathcal{D}}_2$:

$$y \overset{u\cdot a\cdot r}{\sim} \{-1,+1\}, \quad \boldsymbol{x} \sim \mathcal{N}(y\cdot\widehat{\boldsymbol{\mu}}_2,\sigma^2\boldsymbol{I}), \quad \text{where } \widehat{\boldsymbol{\mu}}_2 = (1,3\eta,\ldots,3\eta) \in \mathbb{R}^{d+1}. \tag{10}$$

Theorem 2, restated below, compares the effect of the delusive distributions on natural risk.

**Theorem 2 (restated).** *Let $f_{\mathcal{D}}$, $f_{\widehat{\mathcal{D}}_1}$, and $f_{\widehat{\mathcal{D}}_2}$ be the Bayes optimal classifiers for the mixture-Gaussian distributions $\mathcal{D}$, $\widehat{\mathcal{D}}_1$, and $\widehat{\mathcal{D}}_2$, defined in Eqs. 5, 6, and 7, respectively. For any $\eta > 0$, we have*

$$\mathcal{R}_{\mathsf{nat}}(f_{\mathcal{D}},\mathcal{D}) < \mathcal{R}_{\mathsf{nat}}(f_{\widehat{\mathcal{D}}_2},\mathcal{D}) < \mathcal{R}_{\mathsf{nat}}(f_{\widehat{\mathcal{D}}_1},\mathcal{D}).$$

*Proof.* In the mixture-Gaussian distribution setting described above, the Bayes optimal classifier is linear. In particular, the expression for the classifier of $\mathcal{D}$ is

$$f_{\mathcal{D}}(\boldsymbol{x}) = \arg\max_{c \in \mathcal{Y}} \Pr_{y|\boldsymbol{x}}(y = c) = \operatorname{sign}(\boldsymbol{\mu}^{\top}\boldsymbol{x}).$$

Similarly, the Bayes optimal classifiers for $\widehat{\mathcal{D}}_1$ and $\widehat{\mathcal{D}}_2$ are respectively given by

$$f_{\widehat{\mathcal{D}}_1}(\boldsymbol{x}) = \operatorname{sign}(\widehat{\boldsymbol{\mu}}_1^{\top}\boldsymbol{x}) \quad \text{and} \quad f_{\widehat{\mathcal{D}}_2}(\boldsymbol{x}) = \operatorname{sign}(\widehat{\boldsymbol{\mu}}_2^{\top}\boldsymbol{x}).$$

Now we are ready to calculate the natural risk of each classifier. The natural risk of $f_{\mathcal{D}}(\boldsymbol{x})$ is

$$\begin{aligned}
\mathcal{R}_{\mathsf{nat}}(f_{\mathcal{D}}(\boldsymbol{x}), \mathcal{D}) &= \Pr_{(\boldsymbol{x},y)\sim\mathcal{D}}\left[f_{\mathcal{D}}(\boldsymbol{x}) \neq y\right] \\
&= \Pr_{(\boldsymbol{x},y)\sim\mathcal{D}}\left[\operatorname{sign}(\boldsymbol{\mu}^{\top}\boldsymbol{x}) \neq y\right] \\
&= \Pr\left[y \cdot \left(\mathcal{N}(y,\sigma^2) + \sum_{i=1}^{d}\eta\mathcal{N}(y\eta,\sigma^2)\right) < 0\right] \\
&= \Pr\left[\mathcal{N}(1,\sigma^2) + \sum_{i=1}^{d}\eta\mathcal{N}(\eta,\sigma^2) < 0\right] \\
&= \Pr\left[\mathcal{N}(1 + d\eta^2, (1 + d\eta^2)\sigma^2) < 0\right] \\
&= \Pr\left[\mathcal{N}(0,1) > \frac{\sqrt{1 + d\eta^2}}{\sigma}\right].
\end{aligned}$$

The natural risk of $f_{\widehat{\mathcal{D}}_1}(\boldsymbol{x})$ is

$$\begin{aligned}
\mathcal{R}_{\mathsf{nat}}(f_{\widehat{\mathcal{D}}_1}(\boldsymbol{x}), \mathcal{D}) &= \Pr_{(\boldsymbol{x},y)\sim\mathcal{D}}\left[f_{\widehat{\mathcal{D}}_1}(\boldsymbol{x}) \neq y\right] \\
&= \Pr_{(\boldsymbol{x},y)\sim\mathcal{D}}\left[\operatorname{sign}(\widehat{\boldsymbol{\mu}}_1^{\top}\boldsymbol{x}) \neq y\right] \\
&= \Pr\left[\mathcal{N}(1,\sigma^2) - \sum_{i=1}^{d}\eta\mathcal{N}(\eta,\sigma^2) < 0\right] \\
&= \Pr\left[\mathcal{N}(1 - d\eta^2, (1 + d\eta^2)\sigma^2) < 0\right] \\
&= \Pr\left[\mathcal{N}(0,1) > \frac{1 - d\eta^2}{\sigma\sqrt{1 + d\eta^2}}\right].
\end{aligned}$$

Similarly, the natural risk of $f_{\widehat{\mathcal{D}}_2}(\boldsymbol{x})$ is

$$\begin{aligned}
\mathcal{R}_{\mathsf{nat}}(f_{\widehat{\mathcal{D}}_2}(\boldsymbol{x}), \mathcal{D}) &= \Pr_{(\boldsymbol{x},y)\sim\mathcal{D}}\left[f_{\widehat{\mathcal{D}}_2}(\boldsymbol{x}) \neq y\right] \\
&= \Pr_{(\boldsymbol{x},y)\sim\mathcal{D}}\left[\operatorname{sign}(\widehat{\boldsymbol{\mu}}_2^{\top}\boldsymbol{x}) \neq y\right] \\
&= \Pr\left[\mathcal{N}(1,\sigma^2) + \sum_{i=1}^{d}3\eta\mathcal{N}(\eta,\sigma^2) < 0\right] \\
&= \Pr\left[\mathcal{N}(1 + 3d\eta^2, (1 + 9d\eta^2)\sigma^2) < 0\right] \\
&= \Pr\left[\mathcal{N}(0,1) > \frac{1 + 3d\eta^2}{\sigma\sqrt{1 + 9d\eta^2}}\right].
\end{aligned}$$

Since $\eta > 0$ and $d > 0$, we have $\sqrt{1 + d\eta^2} > \frac{1+3d\eta^2}{\sqrt{1+9d\eta^2}} > \frac{1-d\eta^2}{\sqrt{1+d\eta^2}}$. Therefore, we obtain

$$\mathcal{R}_{\mathsf{nat}}(f_{\mathcal{D}}, \mathcal{D}) < \mathcal{R}_{\mathsf{nat}}(f_{\widehat{\mathcal{D}}_2}, \mathcal{D}) < \mathcal{R}_{\mathsf{nat}}(f_{\widehat{\mathcal{D}}_1}, \mathcal{D}).$$

$\square$

## C.3 Proof of Theorem 3

We consider the problem of minimizing the adversarial risk on some delusive distribution $\widehat{\mathcal{D}}$ by using a linear classifier[11]. Specifically, this can be formulated as:

$$\min_{\boldsymbol{w},b} \mathbb{E}_{(\boldsymbol{x},y)\sim\widehat{\mathcal{D}}} \left[ \max_{\|\boldsymbol{\xi}\|_\infty \le \epsilon} \mathbb{1}\left(\text{sign}(\boldsymbol{w}^\top(\boldsymbol{x}+\boldsymbol{\xi})+b) \ne y\right) \right], \tag{11}$$

where $\mathbb{1}(\cdot)$ is the indicator function and $\epsilon = 2\eta$, the same budget used by the delusive adversary. Denote by $f(\boldsymbol{x}) = \text{sign}(\boldsymbol{w}^\top\boldsymbol{x}+b)$ the linear classifier.

First, we show that the optimal linear $\ell_\infty$ robust classifier for $\widehat{\mathcal{D}}_1$ will rely solely on robust features, similar to the cases in Lemma D.5 of Tsipras et al. [119] and Lemma 2 of Xu et al. [134].

**Lemma 5.** *Minimizing the adversarial risk of the loss (11) on the distribution $\widehat{\mathcal{D}}_1$ (9) results in a classifier that assigns $0$ weight to features $x_i$ for $i \ge 2$.*

*Proof.* The adversarial risk on the distribution $\widehat{\mathcal{D}}_1$ can be written as

$$\begin{aligned}
\mathcal{R}_{\text{adv}}(f,\widehat{\mathcal{D}}_1) &= \Pr\left[\exists \|\boldsymbol{\xi}\|_\infty \le \epsilon, f(\boldsymbol{x}+\boldsymbol{\xi}) \ne y\right]\\
&= \Pr\left[\min_{\|\boldsymbol{\xi}\|_\infty \le \epsilon}(y \cdot f(\boldsymbol{x}+\boldsymbol{\xi})) < 0\right]\\
&= \Pr\left[\max_{\|\boldsymbol{\xi}\|_\infty \le \epsilon}(f(\boldsymbol{x}+\boldsymbol{\xi})) > 0 \mid y=-1\right]\cdot\Pr\left[y=-1\right]\\
&\quad + \Pr\left[\min_{\|\boldsymbol{\xi}\|_\infty \le \epsilon}(f(\boldsymbol{x}+\boldsymbol{\xi})) < 0 \mid y=+1\right]\cdot\Pr\left[y=+1\right]\\
&= \underbrace{\Pr\left[\max_{\|\boldsymbol{\xi}\|_\infty \le \epsilon}\left(w_1(\mathcal{N}(-1,\sigma^2)+\xi_1) + \sum_{i=2}^{d+1} w_i(\mathcal{N}(\eta,\sigma^2)+\xi_i)+b\right) > 0\right]\cdot\frac{1}{2}}_{\mathcal{R}_{\text{adv}}(f,\widehat{\mathcal{D}}_1^{(-1)})}\\
&\quad + \underbrace{\Pr\left[\min_{\|\boldsymbol{\xi}\|_\infty \le \epsilon}\left(w_1(\mathcal{N}(1,\sigma^2)+\xi_1) + \sum_{i=2}^{d+1} w_i(\mathcal{N}(-\eta,\sigma^2)+\xi_i)+b\right) < 0\right]\cdot\frac{1}{2}}_{\mathcal{R}_{\text{adv}}(f,\widehat{\mathcal{D}}_1^{(+1)})}.
\end{aligned}$$

Then we prove the lemma by contradiction. Consider any optimal solution $\boldsymbol{w}$ for which $w_i < 0$ for some $i \ge 2$, we have

$$\mathcal{R}_{\text{adv}}(f,\widehat{\mathcal{D}}_1^{(-1)}) = \Pr\left[\underbrace{\sum_{j \ne i}\max_{|\xi_j| \le \epsilon}\left(w_j(\mathcal{N}(-\widehat{\mu}_{1,j},\sigma^2)+\xi_j)+b\right)}_{\mathbb{A}} + \underbrace{\max_{|\xi_i| \le \epsilon}\left(w_i(\mathcal{N}(\eta,\sigma^2)+\xi_i)\right)}_{\mathbb{B}} > 0\right].$$

Because $w_i < 0$, $\mathbb{B}$ is maximized when $\xi_i = -\epsilon$. Then, the contribution of terms depending on $w_i$ to $\mathbb{B}$ is a normally-distributed random variable with mean $\eta - \epsilon < 0$. Since the mean is negative, setting $w_i$ to zero can only decrease the risk, contradicting the optimality of $\boldsymbol{w}$. Formally,

$$\mathcal{R}_{\text{adv}}(f,\widehat{\mathcal{D}}_1^{(-1)}) = \Pr\left[\mathbb{A} + w_i\mathcal{N}(\eta-\epsilon,\sigma^2) > 0\right] > \Pr\left[\mathbb{A} > 0\right].$$

We can also assume $w_i > 0$ and similar contradiction holds. Similar argument holds for $\mathcal{R}_{\text{adv}}(f,\widehat{\mathcal{D}}_1^{(+1)})$. Therefore, the adversarial risk is minimized when $w_i = 0$ for $i \ge 2$. □

Different from the case in Lemma 5, below we show that the optimal linear $\ell_\infty$ robust classifier for $\widehat{\mathcal{D}}_2$ will rely on both robust and non-robust features.

---

[11]Here we only employing linear classifiers, since considering non-linearity is highly nontrivial for minimizing the $\ell_\infty$ adversarial risk on the mixture-Gaussian distribution [25].

**Lemma 6.** *Minimizing the adversarial risk of the loss (11) on the distribution $\widehat{\mathcal{D}}_2$ (10) results in a classifier that assigns positive weights to features $x_i$ for $i \geq 1$.*

*Proof.* The adversarial risk on the distribution $\widehat{\mathcal{D}}_1$ can be written as

$$\mathcal{R}_{\mathsf{adv}}(f, \widehat{\mathcal{D}}_2) = \underbrace{\Pr\left[\max_{\|\boldsymbol{\xi}\|_\infty \leq \epsilon}\left(w_1(\mathcal{N}(-1, \sigma^2) + \xi_1) + \sum_{i=2}^{d+1} w_i(\mathcal{N}(-3\eta, \sigma^2) + \xi_i) + b\right) > 0\right]}_{\mathcal{R}_{\mathsf{adv}}(f, \widehat{\mathcal{D}}_2^{(-1)})} \cdot \frac{1}{2}$$

$$+ \underbrace{\Pr\left[\min_{\|\boldsymbol{\xi}\|_\infty \leq \epsilon}\left(w_1(\mathcal{N}(1, \sigma^2) + \xi_1) + \sum_{i=2}^{d+1} w_i(\mathcal{N}(3\eta, \sigma^2) + \xi_i) + b\right) < 0\right]}_{\mathcal{R}_{\mathsf{adv}}(f, \widehat{\mathcal{D}}_2^{(+1)})} \cdot \frac{1}{2}.$$

Then we prove the lemma by contradiction. Consider any optimal solution $\boldsymbol{w}$ for which $w_i \leq 0$ for some $i \geq 1$, we have

$$\mathcal{R}_{\mathsf{adv}}(f, \widehat{\mathcal{D}}_2^{(-1)}) = \Pr\left[\underbrace{\sum_{j \neq i} \max_{|\xi_j| \leq \epsilon}\left(w_j(\mathcal{N}(-\widehat{\mu}_{2,j}, \sigma^2) + \xi_j) + b\right)}_{\mathbb{C}} + \underbrace{\max_{|\xi_i| \leq \epsilon}\left(w_i(\mathcal{N}(-\widehat{\mu}_{2,i}, \sigma^2) + \xi_i)\right)}_{\mathbb{D}} > 0\right].$$

Because $w_i \leq 0$, $\mathbb{D}$ is maximized when $\xi_i = -\epsilon$. Then, the contribution of terms depending on $w_i$ to $\mathbb{D}$ is a normally-distributed random variable with mean $-\widehat{\mu}_{2,i} - \epsilon < 0$. Since the mean is negative, setting $w_i$ to be positive can decrease the risk, contradicting the optimality of $\boldsymbol{w}$. Formally,

$$\mathcal{R}_{\mathsf{adv}}(f, \widehat{\mathcal{D}}_2^{(-1)}) = \Pr\left[\mathbb{C} + w_i \mathcal{N}(-\widehat{\mu}_{2,i} - \epsilon, \sigma^2) > 0\right] > \Pr\left[\mathbb{C} + p\mathcal{N}(-\widehat{\mu}_{2,i} - \epsilon, \sigma^2) > 0\right],$$

where $p > 0$ is any positive number. Similar contradiction holds for $\mathcal{R}_{\mathsf{adv}}(f, \widehat{\mathcal{D}}_1^{(+1)})$. Therefore, the optimal solution must assigns positive weights to all features. $\qquad\square$

Now we are ready to derive the optimal linear robust classifiers.

**Lemma 7.** *For the distribution $\widehat{\mathcal{D}}_1$ (9), the optimal linear $\ell_\infty$ robust classifier is*

$$f_{\widehat{\mathcal{D}}_1, \mathsf{rob}}(\boldsymbol{x}) = \mathsf{sign}(\widehat{\boldsymbol{\mu}}_{1, \mathsf{rob}}^\top \boldsymbol{x}), \quad \text{where } \widehat{\boldsymbol{\mu}}_{1, \mathsf{rob}} = (1, 0, \ldots, 0).$$

*Proof.* By Lemma 5, the robust classifier for the distribution $\widehat{\mathcal{D}}_1$ has zero weight on non-robust features (i.e., $w_i = 0$ for $i \geq 2$). Also, the robust classifier will assign positive weight to the robust feature (i.e., $w_1 > 0$). This is similar to the case in Lemma 6 and we omit the proof here. Therefore, the adversarial risk on the distribution $\widehat{\mathcal{D}}_1$ can be simplified by solving the inner maximization problem first. Formally,

$$\begin{aligned}
\mathcal{R}_{\mathsf{adv}}(f, \widehat{\mathcal{D}}_1) &= \Pr\left[\exists \|\boldsymbol{\xi}\|_\infty \leq \epsilon, f(\boldsymbol{x} + \boldsymbol{\xi}) \neq y\right] \\
&= \Pr\left[\min_{\|\boldsymbol{\xi}\|_\infty \leq \epsilon}(y \cdot f(\boldsymbol{x} + \boldsymbol{\xi})) < 0\right] \\
&= \Pr\left[\max_{\|\boldsymbol{\xi}\|_\infty \leq \epsilon}(f(\boldsymbol{x} + \boldsymbol{\xi})) > 0 \mid y = -1\right] \cdot \Pr\left[y = -1\right] \\
&\quad + \Pr\left[\min_{\|\boldsymbol{\xi}\|_\infty \leq \epsilon}(f(\boldsymbol{x} + \boldsymbol{\xi})) < 0 \mid y = +1\right] \cdot \Pr\left[y = +1\right] \\
&= \Pr\left[\max_{\|\boldsymbol{\xi}\|_\infty \leq \epsilon}\left(w_1(\mathcal{N}(-1, \sigma^2) + \xi_1) + b\right) > 0\right] \cdot \Pr\left[y = -1\right] \\
&\quad + \Pr\left[\min_{\|\boldsymbol{\xi}\|_\infty \leq \epsilon}\left(w_1(\mathcal{N}(1, \sigma^2) + \xi_1) + b\right) < 0\right] \cdot \Pr\left[y = +1\right] \\
&= \Pr\left[w_1\mathcal{N}(\epsilon - 1, \sigma^2) + b > 0\right] \cdot \Pr\left[y = -1\right] \\
&\quad + \Pr\left[w_1\mathcal{N}(1 - \epsilon, \sigma^2) + b < 0\right] \cdot \Pr\left[y = +1\right],
\end{aligned}$$

which is equivalent to the natural risk on a mixture-Gaussian distribution $\widehat{\mathcal{D}}_1^*$: $\boldsymbol{x} \sim \mathcal{N}(y \cdot \widehat{\boldsymbol{\mu}}_1^*, \sigma^2 \boldsymbol{I})$, where $\widehat{\boldsymbol{\mu}}_1^* = (1 - \epsilon, 0, \ldots, 0)$. The Bayes optimal classifier for $\widehat{\mathcal{D}}_1^*$ is $f_{\widehat{\mathcal{D}}_1^*}(\boldsymbol{x}) = \text{sign}(\widehat{\boldsymbol{\mu}}_1^{*\top} \boldsymbol{x})$. Specifically,

$$
\begin{aligned}
\mathcal{R}_{\text{nat}}(f, \widehat{\mathcal{D}}_1^*) &= \Pr\left[f(\boldsymbol{x}) \neq y\right] \\
&= \Pr\left[y \cdot f(\boldsymbol{x}) < 0\right] \\
&= \Pr\left[w_1 \mathcal{N}(\epsilon - 1, \sigma^2) + b > 0\right] \cdot \Pr\left[y = -1\right] \\
&\quad + \Pr\left[w_1 \mathcal{N}(1 - \epsilon, \sigma^2) + b < 0\right] \cdot \Pr\left[y = +1\right],
\end{aligned}
$$

which can be minimized when $w_1 = 1 - \epsilon$ and $b = 0$. At the same time, $f_{\widehat{\mathcal{D}}_1^*}(\boldsymbol{x})$ is equivalent to $f_{\widehat{\mathcal{D}}_1, \text{rob}}(\boldsymbol{x})$, since $\text{sign}((1 - \epsilon)x_1) = \text{sign}(x_1)$. This concludes the proof of the lemma.

$\square$

**Lemma 8.** *For the distribution $\widehat{\mathcal{D}}_2$ (10), the optimal linear $\ell_\infty$ robust classifier is*

$$
f_{\widehat{\mathcal{D}}_2, \text{rob}}(\boldsymbol{x}) = \text{sign}(\widehat{\boldsymbol{\mu}}_{2, \text{rob}}^\top \boldsymbol{x}), \quad \text{where } \widehat{\boldsymbol{\mu}}_{2, \text{rob}} = (1 - 2\eta, \eta, \ldots, \eta).
$$

*Proof.* By Lemma 6, the robust classifier for the distribution $\widehat{\mathcal{D}}_2$ has positive weight on all features (i.e., $w_i > 0$ for $i \geq 1$). Therefore, the adversarial risk on the distribution $\widehat{\mathcal{D}}_2$ can be simplified by solving the inner maximization problem first. Formally,

$$
\begin{aligned}
\mathcal{R}_{\text{adv}}(f, \widehat{\mathcal{D}}_2) &= \Pr\left[\exists \, \|\boldsymbol{\xi}\|_\infty \leq \epsilon, f(\boldsymbol{x} + \boldsymbol{\xi}) \neq y\right] \\
&= \Pr\left[\min_{\|\boldsymbol{\xi}\|_\infty \leq \epsilon} (y \cdot f(\boldsymbol{x} + \boldsymbol{\xi})) < 0\right] \\
&= \Pr\left[\max_{\|\boldsymbol{\xi}\|_\infty \leq \epsilon} \left(w_1(\mathcal{N}(-1, \sigma^2) + \xi_1) + \sum_{i=2}^{d+1} w_i(\mathcal{N}(-3\eta, \sigma^2) + \xi_i) + b\right) > 0\right] \cdot \Pr\left[y = -1\right] \\
&\quad + \Pr\left[\min_{\|\boldsymbol{\xi}\|_\infty \leq \epsilon} \left(w_1(\mathcal{N}(1, \sigma^2) + \xi_1) + \sum_{i=2}^{d+1} w_i(\mathcal{N}(3\eta, \sigma^2) + \xi_i) + b\right) < 0\right] \cdot \Pr\left[y = +1\right] \\
&= \Pr\left[\max_{|\xi_1| \leq \epsilon} \left(w_1(\mathcal{N}(-1, \sigma^2) + \xi_1)\right) + \sum_{i=2}^{d+1} \max_{|\xi_i| \leq \epsilon} \left(w_i(\mathcal{N}(-3\eta, \sigma^2) + \xi_i)\right) + b > 0\right] \cdot \Pr\left[y = -1\right] \\
&\quad + \Pr\left[\min_{|\xi_1| \leq \epsilon} \left(w_1(\mathcal{N}(1, \sigma^2) + \xi_1)\right) + \sum_{i=2}^{d+1} \min_{|\xi_i| \leq \epsilon} \left(w_i(\mathcal{N}(3\eta, \sigma^2) + \xi_i)\right) + b < 0\right] \cdot \Pr\left[y = +1\right] \\
&= \Pr\left[w_1 \mathcal{N}(\epsilon - 1, \sigma^2) + \sum_{i=2}^{d+1} w_i \mathcal{N}(\epsilon - 3\eta, \sigma^2) + b > 0\right] \cdot \Pr\left[y = -1\right] \\
&\quad + \Pr\left[w_1 \mathcal{N}(1 - \epsilon, \sigma^2) + \sum_{i=2}^{d+1} w_i \mathcal{N}(3\eta - \epsilon, \sigma^2) + b < 0\right] \cdot \Pr\left[y = +1\right] \\
&= \Pr\left[w_1 \mathcal{N}(2\eta - 1, \sigma^2) + \sum_{i=2}^{d+1} w_i \mathcal{N}(-\eta, \sigma^2) + b > 0\right] \cdot \Pr\left[y = -1\right] \\
&\quad + \Pr\left[w_1 \mathcal{N}(1 - 2\eta, \sigma^2) + \sum_{i=2}^{d+1} w_i \mathcal{N}(\eta, \sigma^2) + b < 0\right] \cdot \Pr\left[y = +1\right],
\end{aligned}
$$

which is equivalent to the natural risk on a mixture-Gaussian distribution $\widehat{\mathcal{D}}_2^*$: $\boldsymbol{x} \sim \mathcal{N}(y \cdot \widehat{\boldsymbol{\mu}}_2^*, \sigma^2 \boldsymbol{I})$, where $\widehat{\boldsymbol{\mu}}_2^* = (1 - 2\eta, \eta, \ldots, \eta)$. The Bayes optimal classifier for $\widehat{\mathcal{D}}_2^*$ is $f_{\widehat{\mathcal{D}}_2^*}(\boldsymbol{x}) = \text{sign}(\widehat{\boldsymbol{\mu}}_2^{*\top} \boldsymbol{x})$.

Specifically,

$$
\begin{aligned}
\mathcal{R}_{\mathsf{nat}}(f, \widehat{\mathcal{D}}_2^*) &= \Pr\left[f(\boldsymbol{x}) \neq y\right] \\
&= \Pr\left[y \cdot f(\boldsymbol{x}) < 0\right] \\
&= \Pr\left[w_1 \mathcal{N}(2\eta - 1, \sigma^2) + \sum_{i=2}^{d+1} w_i \mathcal{N}(-\eta, \sigma^2) + b > 0\right] \cdot \Pr\left[y = -1\right] \\
&\quad + \Pr\left[w_1 \mathcal{N}(1 - 2\eta, \sigma^2) + \sum_{i=2}^{d+1} w_i \mathcal{N}(\eta, \sigma^2) + b < 0\right] \cdot \Pr\left[y = +1\right],
\end{aligned}
$$

which can be minimized when $w_1 = 1 - 2\eta$, $w_i = \eta$ for $i \geq 2$, and $b = 0$. Also, $f_{\widehat{\mathcal{D}}_2^*}(\boldsymbol{x})$ is equivalent to $f_{\widehat{\mathcal{D}}_2,\mathsf{rob}}(\boldsymbol{x})$. This concludes the proof of the lemma.

$\square$

We have established that the optimal linear classifiers $f_{\widehat{\mathcal{D}}_1,\mathsf{rob}}$ and $f_{\widehat{\mathcal{D}}_2,\mathsf{rob}}$ in adversarial training. Now we are ready to compare their natural risks with standard classifiers. Theorem 3, restated below, indicates that adversarial training can mitigate the effect of delusive attacks.

**Theorem 3 (restated).** *Let $f_{\widehat{\mathcal{D}}_1,\mathsf{rob}}$ and $f_{\widehat{\mathcal{D}}_2,\mathsf{rob}}$ be the optimal linear $\ell_\infty$ robust classifiers for the delusive distributions $\widehat{\mathcal{D}}_1$ and $\widehat{\mathcal{D}}_2$, defined in Eqs. 6 and 7, respectively. For any $0 < \eta < 1/3$, we have*

$$
\mathcal{R}_{\mathsf{nat}}(f_{\widehat{\mathcal{D}}_1}, \mathcal{D}) > \mathcal{R}_{\mathsf{nat}}(f_{\widehat{\mathcal{D}}_1,\mathsf{rob}}, \mathcal{D}) \quad and \quad \mathcal{R}_{\mathsf{nat}}(f_{\widehat{\mathcal{D}}_2}, \mathcal{D}) > \mathcal{R}_{\mathsf{nat}}(f_{\widehat{\mathcal{D}}_2,\mathsf{rob}}, \mathcal{D}).
$$

*Proof.* The natural risk of $f_{\widehat{\mathcal{D}}_1,\mathsf{rob}}(\boldsymbol{x})$ is

$$
\begin{aligned}
\mathcal{R}_{\mathsf{nat}}(f_{\widehat{\mathcal{D}}_1,\mathsf{rob}}(\boldsymbol{x}), \mathcal{D}) &= \Pr_{(\boldsymbol{x},y)\sim\mathcal{D}}\left[f_{\widehat{\mathcal{D}}_1,\mathsf{rob}}(\boldsymbol{x}) \neq y\right] \\
&= \Pr_{(\boldsymbol{x},y)\sim\mathcal{D}}\left[\mathrm{sign}(\widehat{\boldsymbol{\mu}}_{1,\mathsf{rob}}^\top \boldsymbol{x}) \neq y\right] \\
&= \Pr\left[\mathcal{N}(1, \sigma^2) < 0\right] \\
&= \Pr\left[\mathcal{N}(0, 1) > \frac{1}{\sigma}\right].
\end{aligned}
$$

Similarly, the natural risk of $f_{\widehat{\mathcal{D}}_2,\mathsf{rob}}(\boldsymbol{x})$ is

$$
\begin{aligned}
\mathcal{R}_{\mathsf{nat}}(f_{\widehat{\mathcal{D}}_2,\mathsf{rob}}(\boldsymbol{x}), \mathcal{D}) &= \Pr_{(\boldsymbol{x},y)\sim\mathcal{D}}\left[f_{\widehat{\mathcal{D}}_2,\mathsf{rob}}(\boldsymbol{x}) \neq y\right] \\
&= \Pr_{(\boldsymbol{x},y)\sim\mathcal{D}}\left[\mathrm{sign}(\widehat{\boldsymbol{\mu}}_{2,\mathsf{rob}}^\top \boldsymbol{x}) \neq y\right] \\
&= \Pr\left[(1 - 2\eta)\mathcal{N}(1, \sigma^2) + \sum_{i=1}^{d} \eta\mathcal{N}(\eta, \sigma^2) < 0\right] \\
&= \Pr\left[\mathcal{N}(1 - 2\eta + d\eta^2, ((1 - 2\eta)^2 + d\eta^2)\sigma^2) < 0\right] \\
&= \Pr\left[\mathcal{N}(0, 1) > \frac{1 - 2\eta + d\eta^2}{\sigma\sqrt{(1 - 2\eta)^2 + d\eta^2}}\right].
\end{aligned}
$$

Recall that the natural risk of the standard classifier $f_{\widehat{\mathcal{D}}_1}(\boldsymbol{x})$ is

$$
\mathcal{R}_{\mathsf{nat}}(f_{\widehat{\mathcal{D}}_1}(\boldsymbol{x}), \mathcal{D}) = \Pr\left[\mathcal{N}(0, 1) > \frac{1 - d\eta^2}{\sigma\sqrt{1 + d\eta^2}}\right],
$$

and the natural risk of the standard classifier $f_{\widehat{\mathcal{D}}_2}(\boldsymbol{x})$ is

$$\mathcal{R}_{\mathsf{nat}}(f_{\widehat{\mathcal{D}}_2}(\boldsymbol{x}), \mathcal{D}) = \Pr\left[\mathcal{N}(0,1) > \frac{1 + 3d\eta^2}{\sigma\sqrt{1 + 9d\eta^2}}\right].$$

It is easy to see that $\frac{1-d\eta^2}{\sqrt{1+d\eta^2}} < 1$. Thus, we have

$$\mathcal{R}_{\mathsf{nat}}(f_{\widehat{\mathcal{D}}_1}, \mathcal{D}) > \mathcal{R}_{\mathsf{nat}}(f_{\widehat{\mathcal{D}}_1,\mathsf{rob}}, \mathcal{D}).$$

Also, $\frac{1+3d\eta^2}{\sqrt{1+9d\eta^2}} < \frac{1-2\eta+d\eta^2}{\sqrt{(1-2\eta)^2+d\eta^2}}$ is true when $0 < \eta < 1/3$ and $d > 0$. Therefore, we obtain

$$\mathcal{R}_{\mathsf{nat}}(f_{\widehat{\mathcal{D}}_2}, \mathcal{D}) > \mathcal{R}_{\mathsf{nat}}(f_{\widehat{\mathcal{D}}_2,\mathsf{rob}}, \mathcal{D}).$$

This concludes the proof.

$\square$

## D    Practical Attacks for Testing Defense (Detailed Version)

In this section, we introduce the attacks used in our experiments to show the destructiveness of delusive attacks on real datasets and thus the necessity of adversarial training.

In practice, we focus on the empirical distribution $\mathcal{D}_n$ over $n$ data points drawn from $\mathcal{D}$. To avoid the difficulty to search through the entire $\infty$-Wasserstein ball, one common choice is to consider the following set of empirical distributions [147]:

$$\mathcal{A}(\mathcal{D}_n, \epsilon) = \left\{ \frac{1}{n} \sum_{(\boldsymbol{x}, y) \sim \mathcal{D}_n} \delta(\boldsymbol{x}', y) : \boldsymbol{x}' \in \mathcal{B}(\boldsymbol{x}, \epsilon) \right\}, \tag{12}$$

where $\delta(\boldsymbol{x}, y)$ is the dirac measure at $(\boldsymbol{x}, y)$. Note that the considered set $\mathcal{A}(\mathcal{D}_n, \epsilon) \subseteq \mathcal{B}_{\mathrm{W}_\infty}(\mathcal{D}_n, \epsilon)$, since each perturbed point $\boldsymbol{x}'$ is at most $\epsilon$-away from $\boldsymbol{x}$.

The L2C attack proposed in Feng et al. [32] is actually searching for the worst-case data in $\mathcal{A}(\mathcal{D}_n, \epsilon)$ with $\ell_\infty$ metric. However, L2C directly optimizes the bi-level optimization problem (4), resulting in a very huge computational cost. Instead, we present five efficient attacks below, which are inspired by "non-robust features suffice for classification" [56]. Our delusive attacks are constructed by injecting non-robust features correlated consistently with a specific label to each example.

**Poison 1** (P1**: Adversarial perturbations**): The first construction is similar to that of the deterministic dataset in Ilyas et al. [56]. In our construction, the robust features are still correlated with their original labels. We modify each input-label pair $(\boldsymbol{x}, y)$ as follows. We first select a target class $t$ deterministically according to the source class $y$ (e.g., using a fixed permutation of labels). Then, we add a small adversarial perturbation to $\boldsymbol{x}$ in order to ensure that it is misclassified as $t$ by a standard model. Formally:

$$\boldsymbol{x}_{\mathsf{adv}} = \underset{\boldsymbol{x}' \in \mathcal{B}(\boldsymbol{x}, \epsilon)}{\arg\min}\ \ell\left(f_{\mathcal{D}}(\boldsymbol{x}'), t\right), \tag{13}$$

where $f_{\mathcal{D}}$ is a standard classifier trained on the original distribution $\mathcal{D}$ (or its finite-sample counterpart $\mathcal{D}_n$). Finally, we assign the correct label $y$ to the perturbed input. The resulting input-label pairs $(\boldsymbol{x}_{\mathsf{adv}}, y)$ make up the delusive dataset $\widehat{\mathcal{D}}_{\mathsf{P1}}$. This attack resembles the mixture-Gaussian distribution $\widehat{\mathcal{D}}_1$ in Eq. (6).

It is worth noting that this type of data poisoning was mentioned in the addendum of Nakkiran [79], but was not gotten further exploration. Concurrently, this attack is also suggested in Fowl et al. [34] and achieves state-of-the-art performance by employing additional techniques such as differentiable data augmentation. Importantly, our P3 attack (a variant of P1) can yield a competitive performance compared with Fowl et al. [34] under $\ell_\infty$ threat model (see Table 2 in our main text).

**Poison 2** (P2**: Hypocritical perturbations**): This attack is motivated by recent studies on so-called "hypocritical examples" [115] or "unadversarial examples" [99]. Here we inject helpful non-robust features to the inputs so that a standard model can easily produce a correct prediction. Formally:

$$\boldsymbol{x}_{\mathsf{hyp}} = \underset{\boldsymbol{x}' \in \mathcal{B}(\boldsymbol{x}, \epsilon)}{\arg\min}\ \ell\left(f_{\mathcal{D}}(\boldsymbol{x}'), y\right). \tag{14}$$

The resulting input-label pairs $(\boldsymbol{x}_{\mathsf{hyp}}, y)$ make up the delusive dataset $\widehat{\mathcal{D}}_{\mathsf{P2}}$, where the helpful features are prevalent in all examples. However, those artificial features are relatively sparse in the clean data. Thus the resulting classifier that overly relies on the artificial features may perform poorly on the clean test set. This attack resembles the mixture-Gaussian distribution $\widehat{\mathcal{D}}_2$ in Eq. (7).

As a note, this attack can be regarded as a special case of the error-minimizing noise proposed in Huang et al. [53], where an alternating iterative optimization process is designed to generate perturbations. In contrast, our P2 attack is a one-level problem that makes use of a pre-trained standard classifier.

**Poison 3** (P3: **Universal adversarial perturbations**): This attack is a variant of P1. To improve the transferability of the perturbation between examples, we adopt the class-specific universal adversarial perturbation [77, 57]. Formally:

$$\boldsymbol{\xi}_t = \underset{\boldsymbol{\xi} \in \mathcal{B}(\mathbf{0}, \epsilon)}{\arg\min} \underset{(\boldsymbol{x}, y) \sim \mathcal{D}}{\mathbb{E}} \ell\left(f_{\mathcal{D}}(\boldsymbol{x} + \boldsymbol{\xi}), t\right), \tag{15}$$

where $t$ is chosen deterministically based on $y$. The resulting input-label pairs $(\boldsymbol{x} + \boldsymbol{\xi}_t, y)$ make up the delusive dataset $\widehat{\mathcal{D}}_{\mathsf{P3}}$. Intuitively, if specific features repeatedly appears in all examples from the same class, the resulting classifier may easily capture such features. Although beyond the scope of this paper, we note that Zhao et al. [144] concurrently find that the class-wise perturbation is also closely related to backdoor attacks.

**Poison 4** (P4: **Universal hypocritical perturbations**): This attack is a variant of P2. We adopt class-specific universal unadversarial perturbations, and the resulting input-label pairs $(\boldsymbol{x} + \boldsymbol{\xi}_y, y)$ make up the delusive dataset $\widehat{\mathcal{D}}_{\mathsf{P4}}$.

**Poison 5** (P5: **Universal random perturbations**): This attack injects class-specific random perturbations to training data. We first generate a random perturbation $\boldsymbol{r}_y \in \mathcal{B}(\mathbf{0}, \epsilon)$ for each class $y$ (using Gaussian noise or uniform noise). Then the resulting input-label pairs $(\boldsymbol{x} + \boldsymbol{r}_y, y)$ make up the delusive dataset $\widehat{\mathcal{D}}_{\mathsf{P5}}$. Despite the simplicity of this attack, we find that it is surprisingly effective in some cases.