# OpenReview forum: "Better Safe Than Sorry: Preventing Delusive Adversaries with Adversarial Training"
_NeurIPS.cc/2021/Conference — NeurIPS 2021 Poster_

### Official Review · Reviewer_5L8s · 2021-07-12

**Rating:** 5
**Confidence:** 4

**Summary:**

The paper proposes a defense method based on adversarial training to defend against delusive adversaries at the training time. The paper shows that adversarial training can largely improve the accuracy under delusive attacks if the defender applies pessimistic budget. Both experiments and theory verify the claims.

**Limitations And Societal Impact:**

See above.

**Main Review:**

**Pros**
1. The delusive attacks are indeed a threat to the training of networks. And adversarial training can largely improve the accuracy under the attacks.
2. The analysis and the theory of the methods are good and reasonable, which point out why adversarial training improves the accuracy.
3. The paper is clearly written and easy to follow.

**Cons**
1. It has been pointed out in [1] (Section 4.3, Under Noise Filtering Techniques) that adversarial training can largely improve the accuracy of delusive attacks. Although that paper does not carefully analyze the phenomenon as this paper, it limits the contribution and novelty of this paper.
2. One biggest drawback of the method is the choices of adversarial budget $\varepsilon$. In Figure 5, the accuracy seems to be sensitive to choice of $\varepsilon$ of the defender. Another question is the generalization across different attack norm. If the attacker chooses attack in $\ell_2$ norm and the defend chooses to defend in $\ell_\infty$ norm, will the accuracy still be improved? The transfer across different attack norm is also a big problem in adversarial training.
3. The analysis is based on the Wasserstein distance. However, the experiments seems to be performed in single $\ell_p$ norm ball, which does not allow different perturbations budget for different training samples.


Overall, I think the paper proposes a good way to prevent delusive attacks. I vote for rejection beacause of
1. Lack of novelty mentioned in point 1 of Cons senction.
2. It also cannot solve more practical problems for different norms in point 2 of Cons section.

I would increase my score if both of the concerns can be resolved.


[1] Hanxun Huang, Xingjun Ma, Sarah Monazam Erfani, James Bailey, Yisen Wang. Unlearnable Examples: Making Personal Data Unexploitable, ICLR 2021.

**Time Spent Reviewing:**

3

---

> ### Author Response · Authors · 2021-08-10
> **Discussion for Reviewer 5L8s on novelty and defender's norms**
>
> Thank you for your constructive feedback, we would like to clarify the novelty of our work and the sensitivity of budgets and norms in the following response. If parts of them are still unclear, please let us know and we are happy to follow up.
>
> &nbsp;
>
> ### Question 1: Clarifications on novelty (e.g., differences from [1])
>
> There might be some misunderstandings of our work. Our contributions are not concealed by [1]. The main contribution of [1] is to propose a kind of delusive attack. They only showed that, in a small part of their experiments, adversarial training *may* be useful as a heuristic remedy for one specific type of attack. However, our main contribution is to show that adversarial training is **theoretically guaranteed** to mitigate various delusive attacks within the budget, no matter what specific method the adversary uses. It is noteworthy that **such a guarantee is crucial,** especially in the context of adversarial machine learning. *Without such a guarantee, one can never be certain that a particular defense mechanism prevents the existence of some delusive attacks.*
>
> We also would like to point out that, in the past few years, the efforts of the community for improving model robustness to test-time adversarial examples have shown that *many heuristic defensive methods could only give a false sense of security*. One lesson that can be learned from the previous efforts is that *it is of great importance to have a good understanding of the guarantees provided*. To this end, our Theorem 1 provides a security guarantee for training-time attacks, in line with the spirit of the guarantee provided by Madry et al. [4] for test-time attacks. Likewise, our understanding for delusive attacks characterized by Theorems 2&3 echoes the mechanism disclosed by Tsipras et al. [5] for adversarial examples. In short, our novel theoretical guarantee and understanding of defending against delusive attacks, along with the thorough experiments, constitute a solid contribution to the adversarial machine learning community.
>
> &nbsp;
>
> ### Question 2: Sensitivity of budgets
>
> The sensitivity of the defender's budget $\epsilon$ is a natural consequence of a defense, and it is not so specific to our work. We note that such sensitivity is very normal for defending against the commonly studied test-time adversarial attacks, where adversarial training with small $\epsilon$ performs poorly on adversarial examples with large perturbations (e.g., see Table "CIFAR10 L2-robust accuracy" at https://github.com/MadryLab/robustness). That is to say, being a defender, one always wants to guarantee the worst-case performance, so it is expected that a small budget is not enough for security.
>
> On evaluating adversarial robustness, it is argued by Carlini et al. [6] that it is necessary to specify a threat model, which includes a set of assumptions about the adversary's goals, capabilities, and budgets. Therefore, in most of our experiments, we considered the typical $\ell_2$ threat model with the budget $\epsilon=0.5$, and both the attacker and the defender comply with the specified budget.
>
> Nevertheless, in order to better understand the interactions between delusive attacks and the proposed defense, we tried our best to explore a more difficult situation where the attacker's budget is unknown to the defender. It turns out that the optimal defender's budget is not hard to be found via a grid search. We also empirically observed that the optimal defender's budget for P1/P2 is different from P3/P4 in Figure 5 (the optimal $\epsilon$ for P1/P2 is roughly 0.25, while the optimal $\epsilon$ for P3/P4 is close to 0.5). A possible explanation for this might be that P1 and P2 are relatively weak and cannot fully utilize the attacker's budget. Note that these attacks are heuristic attacks, which implies that they may not be the global optimal solution of Equation 4. Empirically, P3 and P4 are more optimal (as shown in Figure 4), so it is reasonable that we need a larger budget to defend against them.
>
> Lastly, we also note that such sensitivity may inspire the design of future delusive attacks. A good attack should not only substantially deteriorate the performance of standard training, but also enlarge the defender's budget of adversarial training. We will include this point in our revised manuscript.
>
> &nbsp;
>
> ### Question 3: Generalization across different attack norms
>
> It is a well-known problem that $\ell_{\infty}$ norm based adversarial training fails to be robust against $\ell_2$ norm based adversarial attacks. Fortunately, several orthogonal works have been devoted to addressing this problem, such as [7] and [8]. Using these advanced techniques, one can now adversarially train models that are simultaneously robust to multiple perturbation types of adversarial examples.
>
> In view of the above knowledge, **this actually shows an *advantage* of our theoretical analysis.** In our formulation of delusive attacks, we do not restrict the choice of the distance metric. Therefore, by simply replacing the $\ell_{\infty}$ ball with the union of $\ell_{2}$ ball and $\ell_{\infty}$ ball, our Theorem 1 is still valid. This conclusion implies that we can adversarially train models to simultaneously resist multiple types of delusive attacks (e.g., $\ell_2$ and $\ell_{\infty}$). This advantage has already been (partially) pointed out in line 108 of the main text, and we will make it more clear in our revised manuscript.
>
> &nbsp;
>
> ### Question 4: Different budgets for different samples
>
> Since adversarial training typically applies the same budget $\epsilon$ for all samples, we obey this convention and do not use different $\epsilon$ for different samples. We also would like to point out that the equivalence of adversarial training and distributionally robust optimization is just used as a mathematical tool for deriving our theory. Such equivalence means that in practice, we do not really need to consider the Wasserstein distance, which is complicated and not easy to compute, so we only apply the typical adversarial training methods in experiments.
>
> As a note, several works (such as [9] [10]) have proposed instance adaptive adversarial training to allow different $\epsilon$ for different samples. This direction, while potentially interesting, is *orthogonal* to the problem studied in this paper.
>
> &nbsp;
>
> References:
>
> [1] Hanxun Huang, et al. "Unlearnable Examples: Making Personal Data Unexploitable", ICLR 2021.
>
> [2] Newsome, James, Brad Karp, and Dawn Song. "Paragraph: Thwarting signature learning by training maliciously", International Workshop on Recent Advances in Intrusion Detection, 2006.
>
> [3] Feng, Ji, Qi-Zhi Cai, and Zhi-Hua Zhou. "Learning to Confuse: Generating Training Time Adversarial Data with Auto-Encoder", NeurIPS 2019.
>
> [4] Madry, Aleksander, et al. "Towards deep learning models resistant to adversarial attacks", ICLR 2018.
>
> [5] Tsipras, Dimitris, et al. "Robustness may be at odds with accuracy", ICLR 2019.
>
> [6] Carlini, Nicholas, et al. "On evaluating adversarial robustness", arXiv preprint, 2019
>
> [7] Tramer, Florian, and Dan Boneh. "Adversarial training and robustness for multiple perturbations", NeurIPS 2019.
>
> [8] Maini, Pratyush, Eric Wong, and Zico Kolter. "Adversarial robustness against the union of multiple perturbation models", ICML 2020.
>
> [9] Balaji, Yogesh, Tom Goldstein, and Judy Hoffman. "Instance adaptive adversarial training: Improved accuracy tradeoffs in neural nets", arXiv preprint, 2019.
>
> [10] Ding, Gavin Weiguang, et al. "Mma training: Direct input space margin maximization through adversarial training", ICLR 2020.

---

> ### Author Response · Authors · 2021-08-23
> **Could you please reevaluate our paper based on our clarifications? Thank you!**
>
> Dear Reviewer 5L8s,
>
> Since we are reaching the end of August (and the end of the discussion period), we hope you can re-evaluate our work based on our detailed clarifications.
>
> In short, we have clarified our novelty in the theoretical aspect compared with Huang et al. [1]. We also discussed the sensitivity of budgets and norms of adversarial training.
>
> We believe the low rating is mostly based on the judgment that the existence of [1] may limit the novelty of our work, while the theoretical contributions of our work are solid, which is also recognized by Reviewer LKRg and Reviewer 5bn7. Then we would like to further clarify *the relationship between [1] and our work*.
>
> First of all, we would like to emphasize that ***Huang et al. [1] and our work were independently developed.*** Specifically, *our work was developed **prior to** the appearance of [1] on arXiv*, and the time difference between the two works submitted to arXiv is less than a month. While [1] was a submission of ICLR'21, we actually were not aware of the submission until our work was completed. *If it is necessary, we can provide supporting materials to clarify the independence*, including but not limited to the document history of this work (which has been archived by Overleaf).
>
> Furthermore, ***there are significant differences between the two works.*** The *root cause* of the differences is that *our motivation and contributions are **completely different** from those of [1]*. Please check the following summaries.
>
> 1. ***Our work is a defense paper.*** Our defense is *motivated* by the previous attack papers [2, 3], as indicated by our Introduction section. *However,* ***Huang et al. [1] is an attack paper,*** whose *motivation* is data privacy; as a result, they inadvertently ignore (and do *not* cite) the attack papers [2, 3].
> 2. We are the ***first*** to empirically show that the defense is effective for *various* attacks (including P1, P2, P3, P4, P5, L2C considered in this work), while [1] *only* experiments with adversarial training for their proposed attack (i.e., error-minimizing noises (**E-Min**) in [1]). *Most notably,* ***all of our considered attacks are different from the E-Min attack.*** Therefore, the defense results of the two works are *complementary*. We will revise the related works in our manuscript to reflect that [1] *independently* suggests adversarial training as a heuristic remedy.
> 3. ***Difference between P2 and E-Min.*** We do cite [1] in the attack section (Appendix D) due to some similarity between P2 and E-Min. *However, they are actually different.* Specifically, the E-Min attack in [1] is designed to perform an alternating iterative optimization process, while our P2 attack is a one-level problem. *In fact, our P2 attack is motivated by "unadversarial examples" proposed in [11] and "hypocritical examples" proposed in [12]*, and the optimization of P2 exactly follows [12].
> 4. ***Difference between P1 and E-Max.*** The error-maximizing noises (E-Max) in [1] are *untargeted* adversarial examples, while our P1 attack is constituted by *targeted* adversarial examples. *In fact, our P1 attack is motivated by the deterministic dataset in [13].* Besides, [1] does not experiment with adversarial training for E-Max, while we show that the defense is effective for P1.
> 5. We are the ***first*** to demonstrate that adversarial examples constitute a very effective delusive attack (i.e. the P1 attack), while [1] *fails* to show that their implemented sample-wise adversarial examples are effective (i.e. the E-Max attack).
> 6. We are the ***first*** to point out that the threat model studied in [1, 3] is exactly the *delusive adversary*, which has been studied *ten* years ago in the context of traditional machine learning [2]. Thus, we believe that establishing this connection between the seminal attack paper [2] and the fresh attack papers [1, 3] is one of our contributions.
> 7. We are the ***first*** to provide theoretical analyses of adversarial training for defending against delusive attacks (Theorem 1, 2, and 3 in our work).
>
> *All the above evidences suggest that **there are actually many differences and novel findings in our work.*** *Therefore,* ***our contributions and novelty are not concealed by Huang et al. [1].*** We respectfully hope you can re-evaluate our work based on the clarified relationship between [1] and our work. We believe our contributions are much easier to recognize now.
>
> We sincerely thank you again for your very constructive review, which greatly helped us improve our work. We will be glad to answer any additional questions you may have. Thank you.
>
> Sincerely,
> Paper5226 Authors
>
> &nbsp;
>
> References:
>
> [1] Hanxun Huang, et al. "Unlearnable Examples: Making Personal Data Unexploitable", ICLR 2021.
> [2] Newsome, James, Brad Karp, and Dawn Song. "Paragraph: Thwarting signature learning by training maliciously", RAID, 2006.
> [3] Feng, Ji, Qi-Zhi Cai, and Zhi-Hua Zhou. "Learning to Confuse: Generating Training Time Adversarial Data with Auto-Encoder", NeurIPS 2019.
> [11] Hadi Salman, et al. "Unadversarial examples: Designing objects for robust vision", arXiv preprint, 2020.
> [12] Lue Tao and Songcan Chen. "With false friends like these, who can have self-knowledge?" arXiv preprint, 2020.
> [13] Andrew Ilyas, et al. "Adversarial examples are not bugs, they are features", NeurIPS 2019.

---

> > ### Comment · Reviewer_5L8s · 2021-08-30
> > **Follow-up Comments**
> >
> > Thanks for the detailed response. The response has somewhat addressed my concern about the norm and size of adversarial budget. It is also a problem faced in the test-time adversaries. And many concurrent works are trying to solve the problem.
> >
> > While I admit there are some differences between this work and [1]. The main idea, using adversarial training to defend against delusive attacks, has already been discovered in [1]. The empirical study only verify this conclusion with diverse attacks and thorough evaluations. For the empirical part itself, I still think the novelty is limited.

---

> > > ### Author Response · Authors · 2021-08-31
> > > **Response to follow-up concern about novelty**
> > >
> > > Thank you for your reply.
> > >
> > > While we admit that adversarial training (AT) has been experimented in [1], our main insight is contrary to [1]. Specifically, we have a positive attitude towards AT, but the viewpoint in Huang et al. [1] is somewhat negative: they thought AT could be possibly inactivated. For the empirical part, we have some other interesting insights and findings discovered in this paper for the first time. Please check the following summaries.
> > >
> > > - The most significant insight is that we are the first to consider AT as a *principled defense* against delusive attacks. This claim is supported by theoretical justification (Theorem 1) and understanding (Theorem 2&3) in our paper.
> > >   - It is noteworthy that the viewpoint in [1] is somewhat disappointing: they just found that AT is useful as a heuristic remedy for the noises they proposed; however, they still thought that specific methods could be developed to inactivate AT, and thus they did *not* regard AT as a promising defense.
> > >   - In contrast to [1], we show that AT is actually a very promising defense. Our Theorem 1 indicates that AT is capable of defending against any delusive attacks within the considered threat model, no matter what specific method the adversary uses. This suggests that the defense capability of AT is seriously underestimated by [1].
> > >   - Our experiments also indicate that the performance of AT is worthy of further exploration. Specifically, we find that in practice, small $\epsilon$-based AT and curriculum-based AT (such as FAT) can improve the defense performance. We believe that these interesting findings will inspire future work to further improve the effectiveness of AT.
> > > - The second intriguing insight is that, while AT is commonly believed to be harmful to test accuracy, our theoretical analysis justifies that AT actually can protect test accuracy to a great extent when confronted with delusive adversaries. This makes AT more practical in many real-world applications where test-time attacks are rare and high accuracy on clean test data is required.
> > > - The third insight is that, while filtering out the poisoned examples is usually recommended by popular methods of defending against poisoning attacks, our theoretical analysis justifies that the training examples poisoned by delusive adversaries are still useful, and we can actually defend against the threat without abandoning the poisoned examples.
> > > - The fourth insight is the demonstration of the versatility of AT. For example, we for the first time illustrate the game of delusive attacks and defenses in self-supervised learning, which is of great practical significance. Besides, we are the first to demonstrate that AT can overcome simplicity bias, and we also note that the practical effect of delusive attacks could be attributed to the simplicity bias of neural networks.
> > > - In addition, we are the first to show that adversarial examples can constitute a very effective delusive attack, we are the first to establish the connection between the seminal attack paper [2] and the fresh attack papers [1, 3], and we are the first to outperform the attack proposed by [3] on CIFAR-10 (see Figure 7(a), our P3~P5 attacks are more effective than L2C).
> > >
> > > All in all, our theoretical analysis justifies the potential of AT as a principled defense against delusive attacks, and our empirical study itself also has interesting findings. We believe that the insights provided in this paper will greatly promote the adoption of AT in industry and inspire academic research on AT when confronted with delusive adversaries.
> > >
> > > &nbsp;
> > >
> > > References:
> > >
> > > [1] Hanxun Huang, et al. "Unlearnable Examples: Making Personal Data Unexploitable", ICLR 2021.
> > > [2] Newsome, James, Brad Karp, and Dawn Song. "Paragraph: Thwarting signature learning by training maliciously", RAID, 2006.
> > > [3] Feng, Ji, Qi-Zhi Cai, and Zhi-Hua Zhou. "Learning to Confuse: Generating Training Time Adversarial Data with Auto-Encoder", NeurIPS 2019.

---

### Official Review · Reviewer_5bn7 · 2021-07-13

**Rating:** 8
**Confidence:** 4

**Summary:**

The paper tackles the problem of delusive attacks. It reveals that adversarial training, traditionally considered as one of the best defense methods against adversarial examples, can serve as a powerful and elegant defense against delusive attacks. This expands the defense ability of adversarial training from test-time evasion attacks to training-time availability attacks, making the technique significantly more applicable in practice.

**Limitations And Societal Impact:**

Many limitations and negative aspects are discussed in the paper.

**Main Review:**

### Originality

The paper pioneers the use of adversarial training to mitigate delusive attacks. It shows that adversarial training on perturbed examples minimizes an upper bound of natural risk on clean examples. Theoretical analyses also reveal the internal mechanisms of the defense method. As far as I know, these points have never been theoretically justified before.

### Quality

The paper is technically sound and is a solid contribution to the community. Theoretical results are novel and experimental results are remarkable. Both results indicate that adversarial training is a promising solution to defend against delusive attacks.

### Clarity

The writing of this paper is overall clear and straightforward. The experiments are thorough and well explained.

A few minor observations: line 106: "Eq. (3)" -> "Eq. 3", line 128: "good" -> "a good", line 141: "this case" -> "these cases", line 199: "varying d" -> "vary d", line 140: "changes of" -> "under", line 300: "and, show" -> ", and show", line 354: "optimize" -> "optimizes".



### Significance

1. The paper solves an important and challenging problem of defending against delusive attacks. It provides a novel way to utilize all training data in an elegant manner, rather than abandoning the correctly labeled examples. I believe that many practitioners will be glad to adopt the defense method when their training data is collected from untrusted sources. After all, it's better to be safe than sorry.
2. It is well-known that adversarial training usually leads to a drop in natural accuracy. The paper shows that adversarial training instead can improve the test accuracy on clean examples in the settings of delusive threat models. I found this phenomenon very intriguing. At the same time, this provides a new and convincing argument for the application of adversarial training in many real-world applications.
3. Extensive experiments validate the superiority of the proposed method. The authors also show the versatility of the proposed defense with several experiments; in particular, the authors show the possibility of delusive attacks and defense for rotation-based self-supervised learning. I thought this would inspire researchers to explore the vulnerabilities of SSL and find countermeasures. I also noticed some orthogonal works [1] [2] in this direction that you may find helpful.

### Experiments

1. The authors mention that their implementation of adversarial training minimizes a lower bound of adversarial risk (they adopt the popular PGD-AT method). Why not adopt SOTA certification methods such as CROWN-IBP [3] or Linf-dist Net [4] to directly optimize the adversarial risk?
2. There is still a small gap between the natural accuracy of a standard model trained on the clean data and that of a robust model trained on the perturbed data. Is the gap inevitable?
3. The authors may want to discuss the computation cost of adversarial training compared to standard training.
4. From Figure 5, it turns out that the optimal budgets for P1/P2 and P3/P4 are different. Why is this case? The authors may want to give a reasonable explanation.


[1] Saha et al. Backdoor Attacks on Self-Supervised Learning. 2021

[2] Carlini et al. Poisoning and Backdooring Contrastive Learning. 2021

[3] Zhang et al. Towards stable and efficient training of verifiably robust neural networks. ICLR 2020

[4] Zhang et al. Towards Certifying $\ell_{\infty}$ Robustness using Neural Networks with $\ell_{\infty}$-dist Neurons. ICML 2021

**Time Spent Reviewing:**

8hrs

---

> ### Author Response · Authors · 2021-08-10
> **Discussion for Reviewer 5bn7 on experiments**
>
> Thank you for your supportive review and kind suggestions. We are glad you felt that the paper takes an exciting research direction. Please find our detailed response below.
>
> &nbsp;
>
> ### Question 1: About certified training methods
>
> The current certified training methods, while can minimize an upper bound of the true adversarial risk, have their own drawbacks: 1) their derived upper bounds are usually loose and tend to over-regularize the network during training, resulting in relatively poor performance on the clean test set; 2) their computation costs are higher than PGD-based adversarial training (as they usually require more epochs and more running time per epoch than PGD-based adversarial training). Therefore, we chose the popular PGD-based adversarial training, which is efficient and more suitable for achieving high test accuracy.
>
> &nbsp;
>
> ### Question 2: Is the performance gap inevitable?
>
> It is an intriguing question and in a spirit similar to the tradeoff problem between robustness and accuracy in the context of robustness to test-time adversarial examples. Our theoretical analysis on the mixture-Guassian data showed that the gap is relatively small (or negligible) for $\widehat{\mathcal{D}}_2$ , while the gap is unfortunately inevitable for $\widehat{\mathcal{D}}_1$ even in the limit of infinite data. These findings also corroborate a similar phenomenon observed empirically in more complex settings such as CIFAR-10. We leave it to future work to further reduce the gap.
>
> &nbsp;
>
> ### Question 3: Computation cost of defense
>
> The computation cost of PGD-based adversarial training is roughly 7 times that of ordinary training (we use 7 steps of PGD). However, lots of works have developed fast adversarial training methods (e.g., [1] [2] [3]) with negligible additional cost compared to standard training. Although these techniques are orthogonal to our work, we note that they can be easily adapted to defend against delusive attacks.
>
> &nbsp;
>
> ### Question 4: Optimal budgets for defense
>
> The optimal $\epsilon$ for P1/P2 is 0.25, while the optimal $\epsilon$ for P3/P4 is close to 0.5. A possible explanation for this might be that P1 and P2 are relatively weak attacks and cannot fully utilize the attacker's budget. Note that these attacks are all heuristic attacks, which means that they may not be the global optimal solution of Equation 4. Empirically, P3 and P4 are more effective (as shown in Figure 4), so it is reasonable that we need a larger budget to defend against them.
>
> &nbsp;
>
> References:
>
> [1] Shafahi et al., Adversarial Training for Free! NeurIPS 2019
>
> [2] Zhang et al., You Only Propagate Once: Accelerating Adversarial Training via Maximal Principle, NeurIPS 2019
>
> [3] Wong et al., Fast is better than free: Revisiting adversarial training, ICLR 2020

---

> > ### Comment · Reviewer_5bn7 · 2021-08-30
> > **Novelty concerns regarding this paper**
> >
> > Thanks for your response, which addressed my concerns. However, considering other reviews and responses, it seems that the main criticism about this paper is the novelty of the use of adversarial training, since [1] has experimentally shown that adversarial training can somewhat resist unlearnable examples.
> >
> > Although the authors have clarified the differences between [1] and this paper, I am still wondering about the consequences of these differences. In other words, does this paper provide unique insights that help us better understand the defense against delusive attacks?
> >
> > [1] Huang et al. Unlearnable Examples: Making Personal Data Unexploitable. ICLR 2021.

---

> > > ### Author Response · Authors · 2021-08-31
> > > **Response to follow-up concern about novelty**
> > >
> > > Thank you for your reply.
> > >
> > > In addition to the clear difference from Huang et al. [1], we do have unique and significant insights on adversarial training (AT) as a defense in our paper.
> > >
> > > - The most significant insight is that we are the first to consider AT as a *principled defense* against delusive attacks. This claim is supported by theoretical justification (Theorem 1) and understanding (Theorem 2&3) in our paper.
> > >   - It is noteworthy that the viewpoint in [1] is somewhat disappointing: they just found that AT is useful as a heuristic remedy for the noises they proposed; however, they still thought that specific methods could be developed to invalidate AT, and thus they did *not* regard AT as a promising defense.
> > >   - In contrast to [1], we show that AT is actually a very promising defense. Our Theorem 1 indicates that AT is capable of defending against any delusive attacks within the considered threat model, no matter what specific method the adversary uses. This suggests that the defense capability of AT is seriously underestimated by [1].
> > >   - Our experiments also indicate that the performance of AT is worthy of further exploration. Specifically, we find that in practice, small $\epsilon$-based AT and curriculum-based AT (such as FAT) can improve the defense performance. We believe that these interesting findings will inspire future work to further improve the effectiveness of AT.
> > > - The second intriguing insight is that, while AT is commonly believed to be harmful to test accuracy, our theoretical analysis justifies that AT actually can protect test accuracy to a great extent when confronted with delusive adversaries. This makes AT more practical in many real-world applications where test-time attacks are rare and high accuracy on clean test data is required.
> > > - The third insight is that, while filtering out the poisoned examples is usually recommended by popular methods of defending against poisoning attacks, our theoretical analysis justifies that the training examples poisoned by delusive adversaries are still useful, and we can actually defend against the threat without abandoning the poisoned examples.
> > > - The last insight is the demonstration of the versatility of AT. For example, we for the first time illustrate the game of delusive attacks and defenses in self-supervised learning, which is of great practical significance. Besides, we are the first to demonstrate that AT can overcome simplicity bias, and we also note that the practical effect of delusive attacks could be attributed to the simplicity bias of neural networks.
> > >
> > > All in all, our paper theoretically justifies the potential of AT as a principled defense against delusive attacks. We believe that the insights provided in this paper will greatly promote the adoption of AT in industry and inspire academic research on AT when confronted with delusive adversaries.

---

> > > > ### Comment · Reviewer_5bn7 · 2021-08-31
> > > > **Thank you for the response**
> > > >
> > > > Thanks for the clarification. Now I am confident about the novelty of this paper. I agree that the theoretical justifications are really insightful, and indeed advance the community's understandings of the defense. Thus, I would like to support this paper.

---

> > > > > ### Author Response · Authors · 2021-09-02
> > > > > **Thanks!**
> > > > >
> > > > > We are happy to know that you feel more confident, and we thank you very much for supporting our paper.
> > > > >
> > > > > If there's any outstanding issue where we can improve further, we're happy to invest the time to improve the paper further.
> > > > >
> > > > > Thanks again for your time and consideration.

---

### Official Review · Reviewer_LKRg · 2021-07-15

**Rating:** 7
**Confidence:** 4

**Summary:**

This paper shows adversarial training (AT) can resist delusive attacks naturally.

AT can naturally prevent the learner from overly relying on non-robust features.

**Limitations And Societal Impact:**

See above.

**Main Review:**

1 May I know what the main difference between delusive attacks and poisoning attacks is?

2 A piece of work [1] has shown that adversarial training may intensify the backdoor attack.
Would the claim contradict the finding in [1]?

3 Have authors tried different epsilon_train budgets for adversarial training in defending delusive attacks?

4 AT has a very strong smoothing effect [2], which can smoothen out the non-robust adversarial features.
If AT encounters robust adversarial features, it may strengthen it.

[1] On the trade-off between adversarial and backdoor robustness, in NeurIPS 2020

[2] Understanding the Interaction of Adversarial Training with Noisy Labels.

**Time Spent Reviewing:**

5 hours

---

> ### Author Response · Authors · 2021-08-10
> **Clarifying the contradiction and experiments for Reviewer LKRg**
>
> Thank you for your valuable feedback, we would like to clarify the contradiction and experiments in the following response. If parts of them are still unclear, please let us know and we are happy to follow up.
>
> &nbsp;
>
> ### Question 1: Differences between delusive attacks and poisoning attacks
>
> Generally speaking, data poisoning attacks aim to manipulate the training data to cause a model to fail during inference. The failure modes include misclassification of a particular target image (a.k.a. targeted attacks, or integrity attacks) and misclassification of all data simultaneously (a.k.a. *indiscriminate attacks*, or availability attacks). Therefore, **delusive attacks belong to indiscriminate data poisoning attacks,** as we mentioned in related works in **Appendix E.** Among the indiscriminate attacks, *delusive attacks are the most invisible ones*, since the poisoned examples can maintain their malice even when they are correctly labeled, while other indiscriminate attacks require mislabeling.  We also note that both targeted and indiscriminate attacks were extensively studied for classical models, while for neural networks, most of the existing works focus on targeted misclassification, and there is little work on indiscriminate attacks (e.g., delusive attacks).
>
> &nbsp;
>
> ### Question 2: Would the claim contradict the finding in [1]?
>
> Our claim does *not* contradict the finding in Weng et al. [1]. Specifically, they used experiments to study the interactions between adversarial training and backdoor attacks, which belong to integrity attacks (see Figure 4 in [3] for categorization of attacks). In contrast, the delusive attacks we studied belong to indiscriminate attacks, and both our theoretical and empirical results indicate that adversarial training is a promising solution to defend against delusive attacks.
>
> &nbsp;
>
> ### Question 3: Have authors tried different $\epsilon_{\text{train}}$ budgets?
>
> Yes, we tried. We observed that a budget that is too large may slightly hurt performance, while a budget that is too small is not enough to mitigate the attacks. Empirically, the optimal $\epsilon_{\text{train}}$ for P3 and P4 is close to 0.5, and the optimal $\epsilon_{\text{train}}$ for P1 and P2 is roughly 0.25. See Figure 5 and Section 5.1.3 for more information.
>
> &nbsp;
>
> ### Question 4: On the smoothing effect of AT
>
> Yes, we totally agree with that. Thanks for pointing out the wonderful work, where they showed the smoothing effects of AT under label noise. Orthogonally, part of our analysis (Section 3.2) is based on a similar fact that AT can eliminate the non-robust features caused by delusive adversaries. We will include this study in related works.
>
> &nbsp;
>
> References:
>
> [1] Weng et al., On the Trade-off between Adversarial and Backdoor Robustness, NeurIPS 2020
>
> [2] Zhu et al., Understanding the Interaction of Adversarial Training with Noisy Labels, arXiv preprint, 2021
>
> [3] Biggio and Roli, Wild Patterns: Ten Years After the Rise of Adversarial Machine Learning, Pattern Recognition, 2018

---

> > ### Comment · Reviewer_LKRg · 2021-08-12
> > **### Thanks for the clarifications.**
> >
> > Thank you for the clarification. My concern of the contradiction with [1] is properly solved. It is indeed a different problem setting.
> > Hope the authors could find my reviews useful and update the paper accordingly.
> >
> >
> > The theoretical analysis results in this work are fine for me.
> >
> > After reading the response, I am wondering about the performance of random noise training in your setting (e.g., training with small random noise added to the inputs).
> > Besides, how about leveraging other adversarial training (AT) variants such as instance-dependent based AT and curriculum-based AT for preventing the delusive adversaries.
> > What do authors think about them?

---

> > > ### Author Response · Authors · 2021-08-15
> > > **Thanks for the prompt response and further discussion**
> > >
> > > Thank you for the prompt response and additional thoughtful comments.
> > >
> > > 1. We will include the above discussions in our revised version. They are very helpful and thought-provoking.
> > > 2. Random noise training may not be sufficient to mitigate delusive attacks, since our theoretical analysis suggests to minimize the worst-case risk in the $\epsilon$-ball.
> > > 3. Other AT variants will be effective in our setting because they tackle the worst-case risk.
> > >
> > > To further support this, we consider the suggested instance-dependent-based AT variants (such as MART and GAIRAT) and curriculum-based AT variants (such as CAT, DAT, FAT). Specifically, we chose to experiment with the currently most effective variants among them (i.e., GAIRAT and FAT, according to the latest leaderboard at [RobustBench](https://robustbench.github.io/)). Additionally, we consider random noise training (denoted as RandNoise) using the uniform noise within the $\epsilon$-ball for comparison. We also report the results of standard training (denoted as Standard) and the conventional PGD-based adversarial training (denoted as PGD-AT) for reference.
> > >
> > > The results are summarized below. Here we report the test accuracy of ResNet-18 on the CIFAR-10 dataset under the threat model of delusive adversaries with the $\ell_{\infty}$ ball and $\epsilon=8/255$.
> > >
> > > | Defense \ Attack | L2C       | P1        | P2        | P3        | P4        | P5        |
> > > | ---------------- | --------- | --------- | --------- | --------- | --------- | --------- |
> > > | Standard         | 15.76     | 15.70     | 61.35     | 9.40      | 13.58     | 10.12     |
> > > | RandNoise        | 17.10     | 17.32     | 63.36     | 10.52     | 14.37     | 27.56     |
> > > | PGD-AT           | 82.84     | 84.18     | 86.74     | **86.37** | 83.18     | 84.57     |
> > > | GAIRAT           | 79.96     | 79.61     | 82.68     | 82.05     | 82.81     | 82.28     |
> > > | FAT              | **85.51** | **86.05** | **88.98** | 84.39     | **84.22** | **87.78** |
> > >
> > > As we can see, the performance of random noise training is marginal. In contrast, all AT methods show significant improvements compared with standard training, thanks to the theoretical guarantee provided by our analysis. Besides, we observe that FAT achieves overall better results than other AT variants. This may be due to the tight upper bound of the adversarial risk pursued by FAT. All in all, these new results successfully validate our analysis.
> > >
> > > Hope the above results answer your questions. Please let us know if we can provide further clarification. Thank you again for your time and valuable feedback.

---

> > > > ### Comment · Reviewer_LKRg · 2021-08-18
> > > > **Thanks for the response.**
> > > >
> > > > I appreciate the authors for the clarifications and efforts for the addition experiments.
> > > >
> > > > I am glad to see that an adversarial training variant---FAT---can achieve an overall better performance than others.
> > > > I am happy to see that my suggestions could better shape the paper; authors may update the paper accordingly.
> > > > I am now convinced by the adversarial training for defending the delusive attacks.
> > > >
> > > > Therefore, based on authors's feedback, I would like to increase me scores.
> > > > Adversarial training methods can not only defend adversarial attacks, but also have other values such as denoising incorrect labels and defending delusive attacks (this work).

---

> > > > > ### Author Response · Authors · 2021-08-19
> > > > > **Thanks for your response!**
> > > > >
> > > > > We are happy to know that the concerns about the contradiction and experiments have been well addressed, and we thank the reviewer for engaging with our response and increasing the score. We will incorporate your suggestions in the revised version.
> > > > >
> > > > > If there's any outstanding issue where we can improve further, we're happy to invest the time to improve the paper further.
> > > > >
> > > > > Thanks for your time.

---

### Official Review · Reviewer_ZbSz · 2021-08-02

**Rating:** 6
**Confidence:** 3

**Summary:**

This paper address the problem of adversarial training for delusive attacks with perturbed yet correctly labeled training data. By characterizing delusive adversaries as maximizing the natural risk with perturbed training data (worst-case data) within a specific Wasserstein ball around the original data, the adversarial training (e.g. in [7]) is shown to minimize an upper bound of such adversarial risk (Theorem 1) and hence is principle to defend against such delusive attacks. Furthermore, some settings of perturbing non-robust features are given and discussed to identify several examples of delusive attacks and show how adversarial training is able to achieve lower natural risk than standard models (Theorem 3). To my understanding, the main contribution is the formal analysis of the performance of adversarial training in defending the delusive attacks, with the new metrics employed such as the Wasserstein distance to generate the perturbed data.

**Limitations And Societal Impact:**

The authors have addressed some limitations of the work, but the reviewer has some questions regarding other limitations such as the generality of the adversarial training in classification problems other than the discussed binary classification task with Gaussian data. Authors are suggested to add more discussions to address the comments above.

**Main Review:**

The paper is overall well written in presenting the theoretical analysis of adversarial training against delusive attacks. However, my main concern is in the significance of the contribution in the paper. It seems the main idea is to characterize the delusive attacks using Wasserstein distance as done by [30], and then show the existing adversarial training formulation (e.g. [7]) is also effective in resisting such delusive adversaries by actually minimizing the worst-case adversarial risk. While the results are interesting, I am not sure if such analysis is significantly novel given that no new adversarial training theory nor attacks are proposed to advance the current state of the art adversarial training algorithms, but to verify the performance in different scenarios. To help readers better appreciate the contribution, authors are  encouraged to provide an explicit statement of contribution and specify any actual contributions presented by the theorems.

On the other hand, I have some questions regarding the generality of the adversarial training in resisting against delusive attacks. While Section 3.2 provides some examples and settings to justify how adversarial training will naturally prevent the learner from relying too much on the non-robust features, I was wondering if such training process is able to deal with other classification problems beyond the example of a mixture of two Gaussian distributions? Besides the given binary examples, it would be helpful to give additional discussions on how generalizable the obtained theorem 3 is, e.g. to non-binary classification problems with non-Gaussian data?

Some other questions:

- It is mentioned in the paper that while adversarial training usually would negatively affect its performance on clean data without attacks, the adversarial training using Wasserstein ball is able to achieve good performance even without adversaries. However, I am not able to find explicit discussions to justify this claim. Is this concluded from Theorem 1 that the adversarial training is also minimizing the upper bound of “natural risk” besides the “adversarial risk”, and hence standard training objective is also implied?

- While this paper deals with adversarial training, the reported results in Fig. 2, 5, 6 and 7 do not contain any performance variance information. In order to further justify the results and performance, authors are strongly suggested to provide more quantitative results such as the number of trials and performance variance besides the average results.


**Time Spent Reviewing:**

4

---

> ### Author Response · Authors · 2021-08-10
> **Clarifying the theoretical results and claims for Reviewer ZbSz**
>
> Thank you for your valuable feedback, we would like to clarify our theoretical results and claims in the following response. If parts of them are still unclear, please let us know and we are happy to follow up.
>
> &nbsp;
>
> ### Question 1: Clarifications on our main contributions
>
> We would like to first clear up the main concern which arises from some **factual misunderstandings** of Staib and Jegelka [30]: they *did not characterize delusive attacks* (training-time threats); instead, they *were characterizing adversarial examples* (test-time threats). Besides, we would like to clarify that *what Feng et al. [7] proposed is a delusive attack*, rather than an effective defense.
>
> To our best knowledge, we are the *first* to characterize delusive attacks using Wasserstein distance, and this *novel* characterization paves the way for the derivation of effective defense. We show that, *for the first time*, adversarial training can serve as **a principled defense against delusive attacks with a theoretical guarantee:** adversarial training on the perturbed data is guaranteed to minimize the natural risk on clean data. Therefore, *our theoretical analysis is novel and insightful*, which is also acknowledged by Reviewer 5bn7 and Reviewer 5L8s.
>
> It is also noteworthy that adversarial training, proposed by Goodfellow et al. [9] and reformulated by Madry et al. [19], was only considered by the community as an effective defense method against test-time attacks (e.g., adversarial examples). Therefore, our discovery that adversarial training can reliably resist training-time threats is of great significance.
>
> For better understandings, below we give an explicit statement of our main contributions:
>
> 1. **Formulation of delusive attacks.** We propose *for the first time* to formulate delusive attacks using the $\infty$-Wasserstein distance. This formulation is novel and general, and can cover the formulation of the attack proposed by Feng et al. [7] (i.e. $\mathcal{A} \subseteq \mathcal{B}_{W{\infty}}$ , as mentioned in lines 161-163 in Appendix D). Thus, our defense is guaranteed to defend against [7].
> 2. **The principled defense.** Equipped with the novel characterization of delusive attacks, we are able to get our key contribution: we show that, *for the first time*, adversarial training can serve as a principled defense against delusive attacks with theoretical guarantee (Theorem 1).
> 3. **Internal Mechanisms.** We further disclose the internal mechanisms of the defense in a popular mixture-Gaussian setting (Theorem 2 and Theorem 3).
> 4. **Empirical evidences.** Finally, we complement our theoretical findings with extensive experiments across a wide range of datasets and tasks.
>
> &nbsp;
>
> ### Question 2: Generalization of Theorem 3 to non-Gaussian data
>
> This suggestion brings up interesting points. We totally agree that it is valuable to generalize Theorem 3 to non-Gaussian data, and actually, ***we have empirically shown*** *how generalizable Theorem 2 and Theorem 3 are* to real-world data.
>
> For example, in Section 5.1.1, we have observed that P1 attack is *empirically* more destructive than P2 attack on CIFAR-10 (see lines 267-268). This phenomenon is consistent with Theorem 2, where $\widehat{\mathcal{D}}_1$ is *provably* more harmful than $\widehat{\mathcal{D}}_2$ , noting that P1 and P2 are analogs of $\widehat{\mathcal{D}}_1$ and $\widehat{\mathcal{D}}_2$ (as pointed out in lines 176 and 185 in Appendix D). Furthermore, we have observed that adversarial training *empirically*  improves natural accuracy for all delusive attacks (see lines 265-266). This phenomenon is clearly aligned with Theorem 3.
>
> On the other side, *theoretically generalizing Theorems 2&3 to non-Gaussian data*, while potentially interesting, *is better suited as a future direction*. Two main reasons are elaborated below:
>
> 1. **Gaussian settings are widely used.** *Many theoretical works have adopted such simple binary settings for achieving analytical solutions.* For example, [1], [2], [3], and [4] used similar Gaussian settings to theoretically analyze sample complexity of robust generalization, tradeoff between robustness and accuracy, connections between representation vulnerability and adversarial gap, and unfairness problem of adversarial training algorithms, respectively. *In the spirit of these previous works,* we chose to study internal mechanisms of the proposed defense in the simple yet natural setting.
> 2. **Current analysis is nontrivial.** We would like to note that our derived optimal robust classifier for $\widehat{\mathcal{D}}_2$ is novel, and *there is no similar derivation before*. Also, we stated in Appendix C.3 footnote 10 (below line 108) that considering non-linearity is highly nontrivial for minimizing the adversarial risk, even for the simple mixture-Gaussian distribution.
>
> &nbsp;
>
> ### Question 3: Clarifications on adversarial training
>
> We would like to point out that we *did not* claim "adversarial training using Wasserstein ball is able to achieve good performance even without adversaries".
>
> Instead, we stated the equivalence of "adversarial training" and "distributionally robust optimization (DRO) problem using the $\infty$-Wasserstein distance". Such equivalence implies that in practice, we do not really need to consider the Wasserstein distance, which is complicated and not easy to compute, so we only adopt the typical adversarial training based on point-wise $\epsilon$ ball (i.e., Equation 2) in our experiments.
>
> As a note, our main claim can be seen from Figure 1: adversarial training on the clean training data leads to a drop in natural accuracy (compared with standard training on the clean training data); in contrast, adversarial training on the perturbed training data can significantly improve the natural accuracy (compared with standard training on the perturbed training data). Thus, we claim that adversarial training is "a promising solution to defend against delusive attacks".
>
> &nbsp;
>
> ### Question 4: About performance variance
>
> We chose to maintain the breadth of our experiments over a wide range of attacks, defense budgets, datasets, and tasks. All of the results have *consistently and steadily* demonstrated the effectiveness of the proposed defense, which indicates that, from a broad perspective, the performance variance is small.
>
> To further support the above statement, we repeat five times the experiments in Section 5.1.1. New results are summarized as follows. Here we report mean and standard deviation of the test accuracy for the CIFAR-10 dataset.
>
> | Training method \ Training data | P1         | P2         | P3         | P4         | P5         |
> | ------------------------------- | ---------- | ---------- | ---------- | ---------- | ---------- |
> | Standard training               | 37.87±0.94 | 74.24±1.32 | 15.14±2.10 | 23.69±2.98 | 11.76±0.72 |
> | Adversarial training            | 86.59±0.30 | 89.50±0.21 | 88.12±0.39 | 88.15±0.15 | 88.12±0.43 |
>
> As we can see, the performance deviations of the proposed defense (adversarial training) are very small (< 0.50%), which hardly effect the results.
>
> &nbsp;
>
> References:
>
> [1] Schmidt et al. Adversarially Robust Generalization Requires More Data, NeurIPS 2018
>
> [2] Tsipras et al. Robustness May Be at Odds with Accuracy, ICLR 2019
>
> [3] Zhu et al. Learning Adversarially Robust Representations via Worst-Case Mutual Information Maximization, ICML 2020
>
> [4] Xu et al. To be Robust or to be Fair: Towards Fairness in Adversarial Training, ICML 2021

---

> > ### Comment · Reviewer_ZbSz · 2021-09-01
> > **Thanks for the clarification**
> >
> > Thank you for the detailed responses. I think most of my concerns have been addressed so I am willing to increase my score. In the future version, I would suggest providing a statement of contribution to concisely summarize the novelty and also a more detailed discussion comparing to the existing work (I like the part of Clarifications on our main contributions in the responses). That way it helps readers better understand the connection and differences.

---

> > > ### Author Response · Authors · 2021-09-02
> > > **Thanks!**
> > >
> > > Thank you very much for reading our response and increasing your score.  It is great to hear that the concerns have been addressed and that you appreciated our clarifications on our contributions. We like your suggestion about providing a concise summary of our contributions and a more detailed discussion of previous work, and we will incorporate it in the revised version.
> > >
> > > If there's any outstanding issue where we can improve further, we're happy to invest the time to improve the paper further.
> > >
> > > Thanks again for your time and consideration.

---

> ### Author Response · Authors · 2021-08-23
> **Could you reevaluate our paper based on our clarifications? Thank you.**
>
> Dear Reviewer ZbSz,
>
> We hope you can re-evaluate our paper based on our detailed clarifications, since we are reaching the end of August (and the end of the discussion period).
>
> As you suggested, we have provided an explicit statement of contributions, and also specified actual contributions presented by the theorems. We clarified the generalization ability of the theorems. We also showed that the performance variance is small by repeating the experiments in Section 5.1.1 five times.
>
> We believe the low rating is mostly based on some ***factual misunderstandings*** about our contributions, while the theoretical results of our work are solid, which is also recognized by Reviewer LKRg and Reviewer 5bn7. We respectfully hope you can re-evaluate our paper based on our response to those *misunderstandings* and the explicit statement of our contributions. We believe our contributions are much easier to recognize now.
>
> We really appreciate your very valuable review. Your suggestions have helped us improve our paper a lot. Please feel free to let us know any additional questions you may have. Thank you.
>
> Sincerely,
> Paper5226 Authors

---

### Decision · Program_Chairs · 2021-09-27

**Decision:**

Accept (Poster)

**Comment:**

The paper analyzes in a particular context how adversarial training can help with preventing delusive adversaries affect training. Delusive adversaries are defined to be those that may modify training data (feature, not labels) to make the resulting classifier have bad test error. The analysis is specifically in the case of Gaussian noise. There is also experimental evidence provided. This paper resulted in a long exchange between reviewers and authors and the reviewers turned increasingly positive about the paper. The authors must include all of the additional details that have been included in the discussion in the next iteration of the paper.